# Bioactivity descriptors for uncharacterized chemical compounds

Martino Bertoni [1,6], Miquel Duran-Frigola [1,2,6✉], Pau Badia-i-Mompel [1,6], Eduardo Pauls[1], Modesto Orozco-Ruiz[1], Oriol Guitart-Pla[1], Víctor Alcalde[1], Víctor M. Diaz [3,4], Antoni Berenguer-Llergo [1], Isabelle Brun-Heath [1], Núria Villegas [1], Antonio García de Herreros[3] & Patrick Aloy [1,5✉]

Chemical descriptors encode the physicochemical and structural properties of small molecules, and they are at the core of chemoinformatics. The broad release of bioactivity data has prompted enriched representations of compounds, reaching beyond chemical structures and capturing their known biological properties. Unfortunately, bioactivity descriptors are not available for most small molecules, which limits their applicability to a few thousand well characterized compounds. Here we present a collection of deep neural networks able to infer bioactivity signatures for any compound of interest, even when little or no experimental information is available for them. Our signaturizers relate to bioactivities of 25 different types (including target profiles, cellular response and clinical outcomes) and can be used as drop-in replacements for chemical descriptors in day-to-day chemoinformatics tasks. Indeed, we illustrate how inferred bioactivity signatures are useful to navigate the chemical space in a biologically relevant manner, unveiling higher-order organization in natural product collections, and to enrich mostly uncharacterized chemical libraries for activity against the drug-orphan target Snail1. Moreover, we implement a battery of signature-activity relationship (SigAR) models and show a substantial improvement in performance, with respect to chemistry-based classifiers, across a series of biophysics and physiology activity prediction benchmarks.

[1] Joint IRB-BSC-CRG Programme in Computational Biology, Institute for Research in Biomedicine (IRB Barcelona), The Barcelona Institute of Science and Technology, Barcelona, Catalonia, Spain. [2] Ersilia Open Source Initiative, Cambridge, UK. [3] Programa de Recerca en Càncer, Institut Hospital del Mar d'Investigacions Mèdiques (IMIM) and Departament de Ciències de la Salut, Universitat Pompeu Fabra (UPF), Barcelona, Catalonia, Spain. [4] Faculty of Medicine and Health Sciences, International University of Catalonia, Barcelona, Catalonia, Spain. [5] Institució Catalana de Recerca i Estudis Avançats (ICREA), Barcelona, Catalonia, Spain. [6] These authors contributed equally: Martino Bertoni, Miquel Duran-Frigola, Pau Badia-i-Mompel. ✉email: miquel@ersilia.io; patrick.aloy@irbbarcelona.org

Most of the chemical space remains uncharted and identifying its regions of biological relevance is key to medicinal chemistry and chemical biology[1,2]. To explore and catalog this vast space, scientists have invented a variety of chemical descriptors, which encode physicochemical and structural properties of small molecules. Molecular fingerprints are a widespread form of descriptors consisting of binary (1/0) vectors describing the presence or absence of certain molecular substructures. These encodings are at the core of chemoinformatics and are fundamental in compound similarity searches and clustering, and are applied to computational drug discovery (CDD), structure optimization, and target prediction.

The corpus of bioactivity records available suggests that other numerical representations of molecules are possible, reaching beyond chemical structures and capturing their known biological properties. Indeed, it has been shown that an enriched representation of molecules can be achieved through the use of bioactivity signatures[3]. Bioactivity signatures are multi-dimensional vectors that capture the biological traits of the molecule in a format that is akin to the structural descriptors or fingerprints used in the field of chemoinformatics. The first attempts to develop biological descriptors for chemical compounds encapsulated ligand-binding affinities[4], and fingerprints describing the target profile of small molecules unveiled many unanticipated and physiologically relevant associations[5]. Currently, public databases contain experimentally determined bioactivity data for about a million molecules, which represent only a small percentage of commercially available compounds[6] and a negligible fraction of the synthetically accessible chemical space[7]. In practical terms, this means bioactivity signatures cannot be derived for most compounds, and CDD methods are limited to using chemical information alone as a primary input, thereby hindering their performance and not fully exploiting the bioactivity knowledge produced over the years by the scientific community.

Recently, we integrated the major chemogenomics and drug databases in a single resource named the Chemical Checker (CC), which is the largest collection of small-molecule bioactivity signatures available to date[8]. In the CC, bioactivity signatures are organized by data type (ligand-receptor binding, cell sensitivity profiles, toxicology, etc.), following a chemistry-to-clinics rationale that facilitates the selection of relevant signature classes at each step of the drug discovery pipeline. In essence, the CC is an alternative representation of the small-molecule knowledge deposited in the public domain and, as such, it is also limited by the availability of experimental data and the coverage of its source databases (e.g., ChEMBL[9] or DrugBank[10]). Thus, the CC is most useful when a substantial amount of bioactivity information is available for the molecules and remains of limited value for poorly characterized compounds[11]. In the current study, we present a methodology to infer CC bioactivity signatures for any compound of interest, based on the observation that the different bioactivity spaces are not completely independent, and thus similarities of a given bioactivity type (e.g., targets) can be transferred to other data kinds (e.g., therapeutic indications). Overall, we make bioactivity signatures available for any given compound, assigning confidence to our predictions and illustrating how they can be used to navigate the chemical space in an efficient, biologically relevant manner. Moreover, we explore their added value in the identification of hit compounds against the drug-orphan target Snail1 in a mostly uncharacterized compound library, and through the implementation of a battery of signature–activity relationship (SigAR) models to predict biophysical and physiological properties of molecules.

## Results

The current version of the CC is organized into 5 levels of complexity (A: Chemistry, B: Targets, C: Networks, D: Cells, and E: Clinics), each of which is divided into 5 sublevels (1–5). In total, the CC is composed of 25 spaces capturing the 2D/3D structures of the molecules, targets and metabolic genes, network properties of the targets, cell response profiles, drug indications, and side effects, among others (Fig. 1a). In the CC, each molecule is annotated with multiple n-dimensional vectors (i.e., bioactivity signatures) corresponding to the spaces where experimental information is available. As a result, chemistry (A) signatures are widely available ($\sim 10^6$ compounds), whereas cell-based assays (D) cover about 30,000 molecules, and clinical (E) signatures are known for only a few thousand drugs (Fig. 1b). We thus sought to infer missing signatures for any compound in the CC, based on the observation that the different bioactivity spaces are not completely independent and can be correlated.

Bioactivity signatures must be amenable to similarity calculations, ideally by conventional metrics such as cosine or Euclidean distances, so that short distances between molecule signatures reflect a similar biological behavior. Therefore, inference of bioactivity signatures can be posed as a metric learning problem where observed compound–compound similarities of a given kind are correlated to the full repertoire of CC signatures, so that similarity measures are possible for any compound of interest, including those that are not annotated with experimental data. In practice, for each CC space ($S_i$), we tackle the metric learning problem with a so-called Siamese Neural Network (SNN), having as input a stacked array of CC signatures available for the compound (belonging to any of the A1–E5 layers, $S_1$–$S_{25}$) and as output, an $n$-dimensional embedding optimized to discern between similar and dissimilar molecules in $S_i$. More specifically, we feed the SNN with triplets of molecules (an anchor molecule, one that is similar to the anchor (positive) and one that is not (negative)), and we ask the SNN to correctly classify this pattern with a distance measurement performed in the embedding space (Fig. 1a and Supplementary Fig. 1). We trained 25 such SNNs, corresponding to the 25 spaces available in the CC. We used $10^7$ molecule triplets and chose an SNN embedding dimension of 128 for all CC spaces, scaling it to the norm so as to unify the distance magnitude across SNNs (see "Methods" section for details). As a result of this procedure, we obtained 25 SNN 'signaturizers' ($S_{1-25}$), each of them devoted to one of the CC spaces ($S_i$). A signaturizer takes as input the subset of CC signatures available for a molecule and produces a 128D signature that, in principle, captures the similarity profile of the molecule in the $S_i$ CC space, where experimental information may not be available for the compound.

To handle the acute incompleteness of experimental signatures accessible for training the SNNs (Fig. 1b), we devised a signature-dropout sampling scheme that simulates a realistic prediction scenario where, depending on the CC space of interest ($S_i$), signatures from certain spaces will be available while others may not. For example, in the CC, biological pathway signatures (C3) are directly derived from binding signatures (B4), thus implying that, in a real B4 prediction case, C3 will never serve as a covariate. In practice, signature sampling probabilities for each CC space $S_i$ were determined from the coverage of $S_1$–$S_{25}$ signatures of molecules lacking an experimental $S_i$ signature. Overall, chemical information (A1–5), as well as signatures from large chemogenomics databases (e.g., B4-5), could be used throughout (Supplementary Fig. 2). Signatures related to the subset of drug molecules (e.g., Mode of Action (MoA): B1, indications: E2, side-effects: E3, etc.) were mutually inclusive; however, they were more frequently dropped out in order to extend the applicability of signaturizers beyond the relatively narrow space of known drugs.

We evaluated the performance of a signaturizer $S_i$ in an 80:20 train–test split both (a) as its ability to classify similar and dissimilar compound pairs within the triplets (Fig. 1c and Supplementary Fig. 3), and (b) as the correlation observed between each predicted signature (i.e., obtained without using $S_i$ as part of the input ($S_1$–$S_{25}$)) and, correspondingly, a truth signature

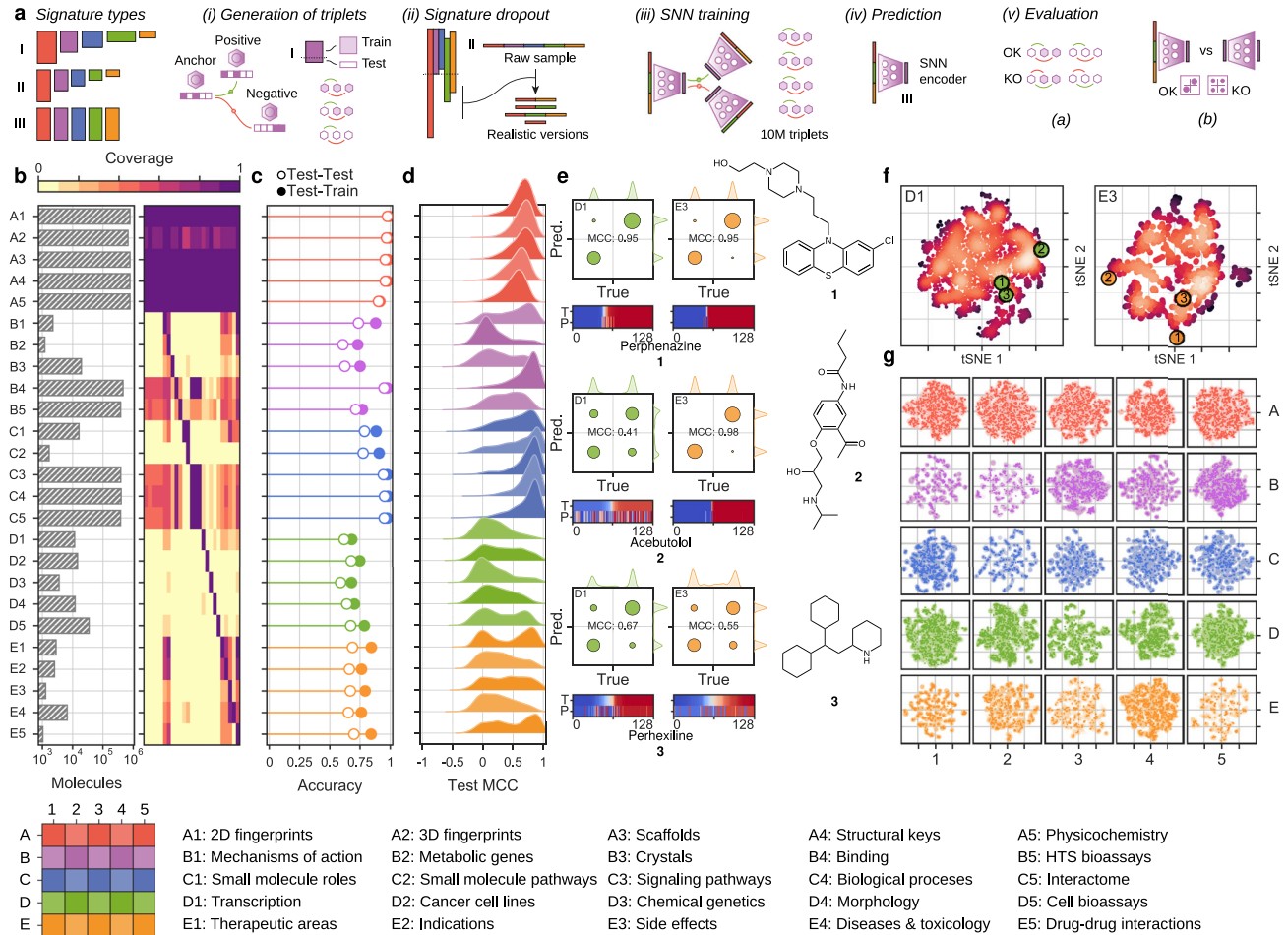

**Fig. 1 Training and evaluation of CC signaturizers. a** Scheme of the methodology. Signaturizers produce bioactivity signatures that fill the gaps in the experimental version of the CC. A SNN is trained using a signature-dropout scheme over $10^7$ triplets of molecules (anchor, positive, negative) to infer missing signatures in each bioactivity space. The inferred signatures are finally evaluated. **b** Coverage of the experimental version of the CC. The bar plot indicates the number of molecules available for each CC data type. The heatmap shows the cross-coverage between data sets, i.e., it is a 25 × 25 matrix capturing the proportion of molecules in one data set (rows) that are also available in other data sets (columns) **c** Accuracy of the 25 signaturizers, measured as the proportion of correctly classified cases within a triplet. Train–test refers to the case where the anchor molecule belongs to the test set, and the positive and negative molecules belong to the training set. Test–test corresponds to the most difficult case where none of the three molecules within the triplet has been utilized during the training. **d** Performance of the 25 signaturizers, measured for each molecule as the correlation between the true and predicted signatures along the 128 dimensions. Given the bimodal distribution of signature values, signatures are binarized (positive/negative) and correlation is measured as a Matthew's correlation coefficient (MCC) over the true-vs-predicted contingency table. **e** Three exemplary molecules (**1**, **2**, and **3**) are shown for the D1 and E3 spaces. True and predicted signatures are displayed as color bars, both sorted according to true signature values. **f** Correspondingly, t-SNE 2D projections of D1 and E3 predictions, where **1**, **2**, and **3** are highlighted, the intensity level describes the density of molecules in the 2D space going from dark red (low density) to white (high density). **g** 2D-projected train (gray) and test (colored) samples for the 25 CC spaces. The legend at the bottom specifies the A1-E5 organization of the CC.

produced using only $S_i$. In the "Methods" section, we further explain these two metrics, as well as the splitting and signature-dropout methods that are key to obtain valid performance estimates. In general, as expected, chemistry (A) signaturizers performed almost perfect (Fig. 1c). The A-level signatures are of little added value, since chemical information is always available for compounds; however, we see that the inferred signatures cannot only faithfully reproduce the original CC signatures but also recapitulate the similarity information captured by the much longer extended connectivity fingerprints (ECFP4; Supplementary Fig. 4). At the targets levels (B), the performance of the signaturizers was high for large-scale binding data B4, while accuracy was variable at deeper annotation levels where the number of compounds available for training was smaller (e.g., MoA (B1) or for drug-metabolizing enzymes (B2)) (Fig. 1d). Performance at the networks level (C) was high, as this level is directly informed

by the underlying targets (B) level. Not surprisingly, the most challenging models were those related to cell-based (D) and clinical (E) data, probably due to the inherent complexity of these data with respect to the number of annotated molecules. On average, the accuracy of cell-based signaturizers was moderate (~0.7) and true-vs-predicted correlation of clinical signatures such as therapeutic classes (Anatomical Therapeutic Chemical, ATC; E1) was variable across molecules. The performance of SNNs varied depending on the CC space and molecule of interest, with signatures being well predicted in all spaces. Figure 1e and f illustrate this observation for three drugs (namely perphenazine (**1**), acebutolol (**2**), and perhexiline (**3**)), which have predicted signatures of variable quality in the transcriptional (D1) and side-effects (E3) spaces. Overall, bioactivity maps were well covered by test-set molecules, indicating that our SNNs are unbiased and able to generate predictions that are spread throughout the

complete bioactivity landscape (Fig. 1g and Supplementary Fig. 5).

**Large-scale inference of bioactivity signatures**. Having trained and validated the signaturizers, we massively inferred missing signatures for the ~800,000 molecules available in the CC, obtaining a complete set of $25 \times 128$-dimensional signatures for each molecule (https://chemicalchecker.org/downloads). To explore the reliability of the inferred signatures, we assigned an applicability score ($\alpha$) to predictions based on the following: (a) the proximity of a predicted signature to true (experimental) signatures available in the training set; (b) the robustness of the SNN output to a test-time data dropout[12]; and (c) the accuracy expected a priori based on the experimental CC data sets available for the molecule (Fig. 2a). A deeper explanation of this score can be found in the "Methods" section, along with Supplementary Fig. 6 showing the relative contribution of a, b, and c factors to the value of $\alpha$. In a similarity search exercise, we found that $\alpha$ scores $\geq 0.5$ retrieved a significant number of true hits (odds-ratios > 8, $P$-values $< 1.7 \cdot 10^{-21}$ (Supplementary Fig. 7)). This observation shows that, even for modest-quality CC spaces such as D1 (transcription), the number of signatures available can be substantially increased by our method (in this case from 11,638 molecules covered in the experimental version of the CC to 69,532 (498% increase) when SNN predictions are included (Supplementary Fig. 8)). Moreover, low- and high-$\alpha$ areas of the signature landscape can be easily delimited, indicating the presence of reliable regions in the prediction space (Fig. 2b).

The $5 \times 5$ organization of the CC (A1–E5) was designed to capture distinct aspects of the chemistry and biology of compounds, and a systematic assessment of the original (experimental) resource revealed partial correlations between the 25 data types[8]. The

original pattern of correlations was preserved among inferred signatures, especially for the high-$\alpha$ ones (Fig. 2c and Supplementary Fig. 9), thereby suggesting that the data integration performed by the SNNs conserves the genuine information contained within each data type, and implying that signatures can be stacked to provide non-redundant, information-rich representations of the molecules. For example, the 25 CC spaces can be concatenated horizontally to obtain a global signature (GSig) of 3200 dimensions ($25 \times 128$D), encapsulating in a unique signature all the bioactivities assigned to a molecule (Fig. 2d). Similarity measures performed in the GSig space up-rank pairs of compounds with the same MoA or ATC code (Fig. 2e) and have an overall correlation with the rest of experimental data available from the CC, capturing not only chemical similarities between molecules but also common target profiles, clinical characteristics and, to a lesser degree, cell-based assay read-outs (Fig. 2f).

Indeed, as shown in Fig. 2g, a 2D projection of GSigs reveals clusters of molecules with specific biological traits. Of note, some of the clusters group molecules with similar chemistries (e.g., ESR1,2 ligands), while others correspond to sets of diverse compounds (e.g., MAPK8,9,10 inhibitors). Most of the clusters have a mixed composition, containing subgroups of chemically related compounds while also including distinct molecules, as is the case for the HSP90AA1-associated cluster, of which compounds **4** and **5** are good representatives (Fig. 2h).

**Bioactivity-guided navigation of the chemical space**. Taken together, CC signatures offer a novel bioactivity-driven means to organize chemical space, with the potential to unveil higher levels of organization that may not be apparent in the light of chemical information alone. In Fig. 3a, we analyze a diverse set of over 30

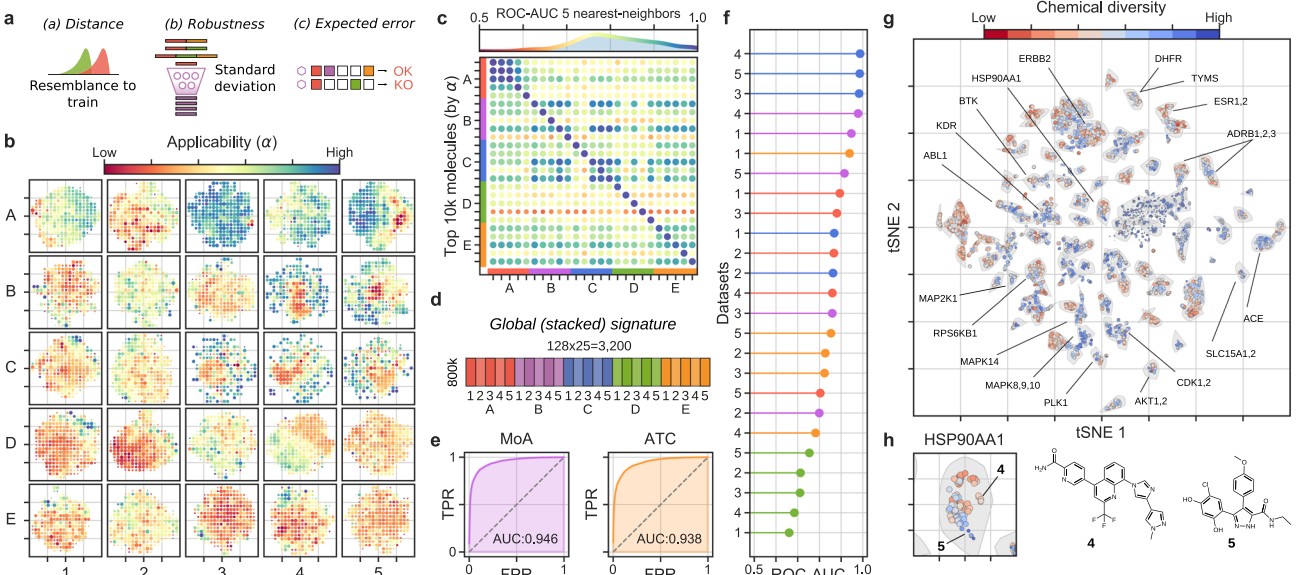

**Fig. 2 Large-scale bioactivity prediction using the signaturizers (~800k molecules). a** Features combined to derive the applicability scores ($\alpha$). **b** Applicability scores for the predictions, displayed across the 25 (A1-E5) 2D-projected signature maps. A grid was defined on the 2D coordinates, molecules were binned and the average $\alpha$ is plotted in a red (low) to blue (high) color scale. **c** Cross-correlation between CC spaces, defined as the capacity of similarities measured in $S_i$ (rows) to recall the top-5 nearest neighbors in $S_j$ (columns) (ROC-AUC), the color scale goes from red to blue indicating low to high cross-correlation (also reported as dot size). Top 10k molecules (sorted by $\alpha$) were chosen as $S_i$. **d** Scheme of the signature stacking procedure. Signatures can be stacked horizontally to obtain a global signature (GSig) of 3200 dimensions. **e** Ability of similarity measures performed in the GSig space to identify pairs of molecules sharing the Mode of action (MoA left) or therapeutic classes (ATC code right) (ROC-AUC). **f** Likewise, the ability of GSigs to identify the nearest neighbors found in the experimental (original) versions of the A1-E5 data sets. **g** t-SNE 2D projection of GSigs. The 10k molecules with the highest average $\alpha$ across the 25 signatures are displayed. The cool-warm color scale represents chemical diversity, red meaning that molecules in the neighborhood are structurally similar (Tanimoto MFp similarity between the molecule in question and their 5-nearest neighbors). A subset of representative clusters is annotated with enriched binding activities. **h** Example of a cluster enriched in heat shock protein 90 inhibitors (HSP90AA1) with highlighted representative molecules with distinct (**4**) or chemically related (**5**) neighbors in the cluster.

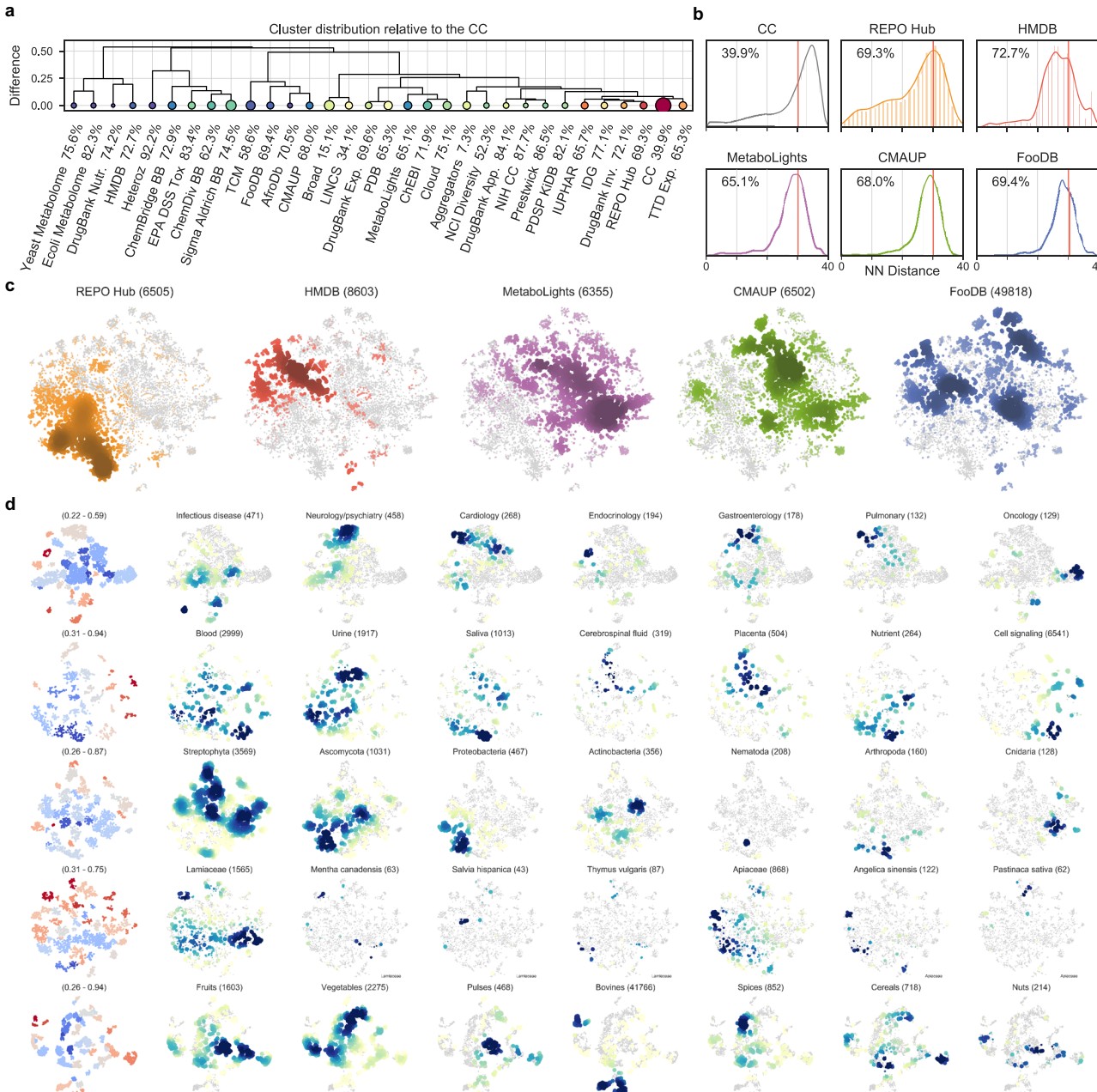

**Fig. 3 Signature-based analysis of compound collections. a** Chemical libraries are hierarchically clustered by their proximity to the full CC; here, proximity is determined by the cluster occupancy vector relative to the k-means clusters identified in the CC collection (number of clusters = $(N/2)^{1/2}$; GSigs are used). Proximal libraries have small Euclidean distances between their normalized occupancy vectors. Size of the circles is proportional to the number of molecules available in the collection. Color (blue-to-red) indicates the homogeneity (Gini coefficient) of the occupancy vectors relative to the CC. **b** Occupancy of high-applicability regions is further analyzed for five collections (plus the full CC). In particular, we measure the average 10-nearest-neighbor L2-distance (measured in the GSig space) of molecules to the high-$\alpha$ subset of CC molecules ($10^3$, Fig. 2). The red line denotes the distance corresponding to an empirical similarity $P$-value of 0.01. The percentage indicates the number of molecules in the collection having high-$\alpha$ vicinities that are, on average, below the significance threshold. This percentage is shown for the rest of the libraries in **a**. **c** The previous five compound collections are merged and projected together (t-SNE). Each of them is highlighted in a different color with darker color indicating a higher density of molecules. **d** Detail of the compound collections. The first column shows the chemical diversity of the projections, measured as the average Tanimoto similarity of the 5-nearest neighbors. Blue denotes high diversity and red high structural similarity between neighboring compounds. Coloring is done on a per-cluster basis. The rest of the columns focus on annotated subsets of molecules. Blue indicates high-density regions.

compound collections, ranging from species-specific metabolomes to purchasable building-block (BB) libraries. To expose the regions of the global bioactivity space covered by these collections, we first performed a large-scale GSig-clustering on the full CC. We then calculated GSigs for each compound in each library and mapped them to the CC clusters, thereby obtaining a specific

cluster occupancy vector for each collection. Finally, we used these vectors to hierarchically group all the compound libraries. As can be seen, drug-related libraries (e.g., IUPHAR and IDG) had similar occupancy vectors to the reference CC library, meaning they were evenly distributed in the bioactivity space, which is expected given the over-representation of medicinal

chemistry in our resource. Libraries containing BBs from different providers (ChemDiv, Sigma-Aldrich, and ChemBridge) were grouped together, although with an uneven representativity of the CC bioactivity space. Similar trends were observed for species-specific metabolomes (Yeast, *E. coli*, and Human (HMDB)) and natural products collected from various sources (Traditional Chinese Medicines (TCM), African substances (AfroDb), or food ingredients (FooDB)).

To gain a better understanding of the bioactivity areas encompassed by each collection, we chose five examples related to drug molecules, metabolomes, and natural product extracts. More specifically, we considered 6505 approved and experimental drugs (REPO Hub)[13], 8603 endogenous human metabolites (HMDB)[14], 6355 metabolites found in other species beyond vertebrates (MetaboLights)[15], 49,818 food constituents (FooDB; www.foodb.ca) and 6502 plant chemicals (CMAUP)[16]. Figure 3b shows that, despite their variable depth of annotation (Supplementary Fig. 10), these collections, for the most part, are laid out in high-$\alpha$ regions of the GSig space. Moreover, Fig. 3c offers a comparative view of the bioactivity areas occupied by each collection, with some overlapping regions as expected, especially between natural product collections. The map reveals a region that is specific to drug molecules, possibly belonging to a set of bioactivities that is outside the reach of natural metabolites.

A deeper dive reveals further structure in the bioactivity maps. For example, when we focus on drug molecules (REPO Hub), broad therapeutic areas such as infectious diseases, neurology/psychiatry, cardiology, and oncology can be circumscribed within certain regions of the GSig landscape (Fig. 3d), and the same applies to finer-grained disease categories (indications) and mechanisms of action (Supplementary Fig. 11). Thus, the chemistry-to-clinics scope of GSigs provides a multi-level view of the chemical space, clustering compounds first on the basis of their targets and, in turn, keeping targets close in space if they belong to the same disease area. This is exemplified by PI3K, CDK, and VEGFR inhibitors, which have their own well-defined clusters within the oncology region of the map, and by histamine receptor antagonists and acetylcholine receptor agonists, which are placed together in an area assigned to neurology/psychiatry (Fig. 3d and Supplementary Fig. 11).

Analogous observations can be made beyond the well-annotated universe of drug molecules, consistently organizing the chemical space in relevant ways. For example, the HMDB map highlights tissue- and biofluid-specific regions with varying degrees of chemical diversity (Fig. 3d and Supplementary Fig. 12), and the MetaboLights cross-species metabolome database is well organized by taxonomy (e.g., Chordata, Ascomycota, Actinobacteria), revealing conserved metabolite regions as well as species-specific ones (in general, we found the former to be less chemically diverse (Supplementary Fig. 12)). Likewise, plants can be organized in families and species by means of their ingredient signatures, as exemplified in Fig. 3d for three *Lamiaceae* and two *Apiaceae* species. Finally, the map of food ingredients displays clear bioactivity clusters of food chemicals, adding to recent work suggesting that the food constituents landscape can be charted and exploited to identify links between diet and health[17].

**Enriching chemical libraries for activity against Snail1.** After seeing that inferred CC signatures are indeed useful to characterize large natural product collections, we sought to assess whether they are also advantageous in combination with more classical chemo-centric approaches. To this end, we performed a computational assessment of two chemical libraries, namely the Prestwick collection (PWCK) and the IRB Barcelona proprietary library (IRB). The IRB library contains >17,000 compounds, only

3% of which have reported bioactivities and are thus included in the CC. This library was originally designed to inhibit t-RNA synthetases by means of ambivalent small molecules displaying ATP-like and amino acid-like chemotypes. The PWCK library is considerably smaller (>1,000 compounds), and it is composed of well-annotated molecules over a wide range of activities (>99% of the molecules are present in the CC). Thus, the IRB and PWCK libraries represent two typical scenarios: the recycling of a targeted library, and the use of a small diversity-oriented compound collection, respectively.

We sought to enrich these libraries for activity against the product of *SNAI1* gene, Snail1, a zinc-finger transcription factor with an essential role in the epithelial-to-mesenchymal transition (EMT)[18]. Being a transcription factor, Snail1 is almost undruggable[19], and we looked for indirect strategies to inhibit its function. In a previous siRNA screening, we found that the knock-down of certain deubiquitinases (DUBs) significantly decreased Snail1 levels, suggesting that DUBs promote Snail1 stabilization and are required for its effects on EMT and cancer progression[20].

We searched the literature for previous knowledge on DUB inhibition by small molecules[21–23] and categorized DUBs on the basis of their performance in the siRNA-DUB/Snail1 screening assay (Supplementary Data 1). We curated 45 DUB inhibitors, 6 of which were inhibitors of candidate DUBs in the siRNA-DUB/Snail1 assay. In parallel, we collected 5540 compound-DUB interactions available in the CC corresponding to 15 of the DUBs. Overall, this search yielded a substantial pool of chemical matter related to DUB inhibition (Supplementary Data 1).

In addition to DUBs, we considered other proteins with a well-established connection to Snail1 activity, including TGFBR1/2, ERK2, FBXL5/14, DDR2, and GSK3B[24]. We collected perturbational (e.g., shRNA) expression signatures for the corresponding genes, together with the signatures of prominent DUBs found in the siRNA-DUB/Snail1 screen. In total, we retrieved 95 transcriptional signatures from the L1000 Connectivity Map and 18 from the Gene Expression Omnibus (GEO)[25] (see Supplementary Data 1 for the full list of signatures). Each signature was converted to the CC D1 format. Finally, we derived networks-level (C) signatures for the previous Snail1-related proteins by exploring their pathways (C3), biological processes (C4), and interactome neighborhoods (C5).

We then devised a strategy to select a few hundred compounds enriched for activity against Snail1 from the IRB and PWCK libraries (Fig. 4a). On the one hand, we defined a chemical query to identify compounds that were (i) chemically similar ($P < 0.001$) to well-curated DUB inhibitors, or to DUB inhibitors in a broader list (combined with binding data from chemogenomics resources). On the other hand, we designed two biological queries to capture connectivities between the biology of Snail1 and the bioactivity data available in the CC. In particular, we looked for (ii) compounds whose (putative) targets were functionally related to Snail1 (i.e., C3–5 similarities to TGFBR1/2, ERK2, etc., $P < 0.001$) but different from DUBs, and (iii) compounds whose gene expression pattern might mimic the transcriptional signatures of genetic KO perturbations of the above targets (i.e., D1 similarities, $P < 0.001$). A detailed description of the queries is given in the "Methods" section.

After inferring CC bioactivity signatures for all the ~20,000 compounds in our libraries, the chemical and biological queries detailed above retrieved 169 and 131 compounds, respectively, with 78 molecules being picked by both. Thus, overall, we selected 222 compounds from the three queries (Supplementary Data 1); 131 from the IRB library and 91 to the PWCK collection. In addition, we selected 188 random compounds to be used as background, using the same library proportions. Selected compounds had comparable molecular weights and drug-like properties (Supplementary Fig. 13a). As expected, compounds identified by the

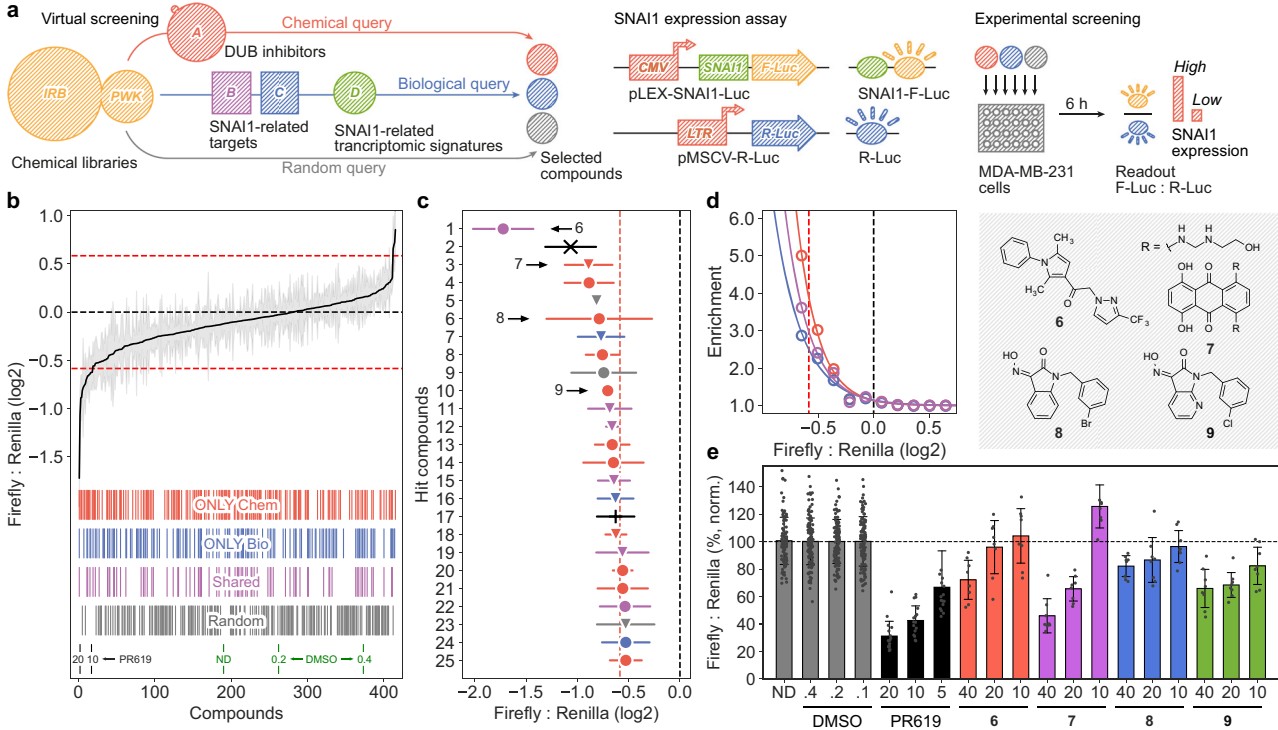

**Fig. 4 Library enrichment to identify Snail1 inhibitors. a** Scheme of the methodology. Two compound libraries are screened (IRB and PWCK). A chemical query is done by looking for similarities with known DUB inhibitors. A biological query is done by looking for transcriptional (D1) and network-based (C3–5) signature matchings with Snail1-relevant targets. Random molecules are selected to estimate the background hit-rate. A Snail1 expression assay based on Firefly:Renilla luciferase ratios are used to screen candidate compounds. **b** Library enrichment quantification showing the effects of compounds selected by chemical (red), biological (blue), shared between both (magenta), and random (gray) queries, as well as the positive (PR-619) and negative (DMSO) controls. **c** Detail of the top 25 hit compounds. **d** Fold enrichment of compounds selected by chemical (red), biological (blue), and shared (magenta) queries with respect to random picks, based on their capacity to modulate Snail1 levels (Firefly:Renilla assay). Median ± MAD ($n = 4$). **e** MDA-MB-231 cells stably expressing luciferase constructs were treated for 6 h with the indicated compounds, at different doses. Firefly:Renilla ratios were normalized with the corresponding concentration of vehicle (DMSO). Mean ± SD of 2 independent experiments, each of them including 4 replicas, are shown.

chemical query are more similar to our reference set of 45 known DUB inhibitors than those retrieved by the biological queries (Supplementary Fig. 13b).

To validate the capacity of these compounds to decrease Snail1 protein levels, we used a Snail1-Firefly-luciferase fusion protein stably expressed in MDA-MB-231 cells (Fig. 4a)[20]. Figure 4b shows the outcome of the Snail1-luciferase screening assay. As can be seen, 22 out of the 25 compounds displaying the strongest Snail1 down-regulation (including the two controls) came from chemical and biological queries. Importantly, a substantial number of hits (6 in the top 25) were candidate molecules selected by both biological and chemical queries, and an additional 3 compounds were retrieved only by biological queries (Supplementary Fig. 13c). It is important to note that these three compounds are chemically unrelated to any of the known DUB inhibitors, with Tanimoto similarities ranging from 0.13 to 0.32 (Supplementary Data 1). Overall, our results highlight the added value of bioactivity signatures to complement chemical similarity searches (Fig. 4c). Certainly, considering as positive those molecules able to decrease 1.5 times Snail1 levels, selected compounds showed a 5-fold enrichment over the hit-rate of random compounds (Fig. 4d). It is also worth noting that 17 of the positive hits were not known to be bioactive, and therefore their CC signatures have been fully inferred by our signaturizers. Finally, we selected the 10 compounds that displayed the strongest effect on reducing Snail1 levels and re-tested them in a confirmatory dose-response assay (Supplementary Fig. 14). Indeed, 4 of them showed a dose-dependent regulation of Snail1

(Fig. 4e). Of note, compounds **8** and **9** had the same chemotype, which was identified in 4 of the top 25 hits. Taken together, these results demonstrate that the various kinds of inferred chemical and biological signatures can be used to implement complex searches to tackle the activity of currently orphan targets.

**Enhanced prediction capabilities compared to chemical descriptors**. In addition, we examined whether our signaturizers could be used as molecular features to predict the outcome of a given bioassay of interest, analogous to the use of chemical descriptors in structure-activity relationship (SAR) studies. We thus developed signature–activity relationship (SigAR) models, and trained machine-learning classifiers to learn discriminative features from the CC signatures of active (1) and inactive (0) compounds, with the goal of assigning a 1/0 label to new (untested) compounds.

To evaluate the SigAR approach in a wide range of scenarios, we used nine state-of-the-art biophysics and physiology benchmark data sets available from MoleculeNet[26]. More specifically, we considered bioassays extracted from PubChem (PCBA), namely an unbiased virtual screening data set (MUV), inhibition of HIV replication (HIV), inhibition of beta-secretase 1 activity (BACE), blood-brain barrier penetration data (BBBP), toxicity experiments (Tox21 and ToxCast), organ-level side effects (SIDER), and clinical trial failures due to safety issues (ClinTox). Although none of these benchmark data sets are explicitly included in the CC resource, data points can be shared between

MoleculeNet and the CC, which would trivialize predictions. To rule out this possibility, we excluded certain CC signature classes from some of the exercises, as detailed in Supplementary Table 1 (e.g., side-effect signatures (E3) were not used in the SIDER set of MoleculeNet tasks).

Each MoleculeNet benchmark data set has a given number of prediction tasks, ranging from 617 (ToxCast) to just one (HIV, BACE, and BBBP). The number of molecules also varies (from 1,427 in SIDER to 437,929 in PCBA) (Supplementary Table 1). We trained a classifier for each MoleculeNet task independently, following a conformal prediction scheme that relates the prediction score to a measure of confidence[27]. We chose to use a general-purpose machine-learning method (i.e., a random forest classifier) with automated hyperparameter tuning, allowing us to focus on the added value of the CC signatures rather than the classification algorithm. Nevertheless, to confirm that the observed trends are not dependent on the random forest classifier, we repeated the experiment with a model-agnostic approach based on an AutoML methodology. To evaluate the accuracy of the classifiers, we followed MoleculeNet recommendations strictly, both in terms of splitting (e.g., scaffold-based) and in terms of performance measure (AUROC/AUPR) to ensure a fair assessment. We primarily compared the performance of CC signatures with the popular Morgan fingerprint (MFp), but also included a collection of different chemical descriptors such as MACCS keys, Daylight-like, and continuous and data-driven descriptors. We also included in the comparison the performance of the best predictor for each task as reported in MoleculeNet. Finally, although CC signatures are abstract representations that do not offer direct structural/mechanistic interpretations, we devised a strategy to obtain high-level explanations for predicted activities. More specifically, for each molecule, we measured the cumulative explanatory potential (Shapley values[28]) of each signature type ($S_{1-25}$) across the GSig space, indicating the classes of data (chemistry, targets, etc.) that were more determinant for the classifier decision (see "Methods" section). In sum, we implemented an automated (parameter-free) SigAR methodology, the outcome of which can be interpreted at the signature-type level and is calibrated as a probability or confidence score.

In Fig. 5a–d and Supplementary Fig. 15, we show the characteristics of a representative classifier, corresponding to the heat shock factor response element (SR-HSE) task in the Tox21 panel. In a 5-fold cross-validation, active molecules got higher prediction scores than inactive compounds (Supplementary Fig. 15). Moreover, the SigAR model outperformed the conventional chemical MFp (Fig. 5a).

Additionally, the accuracy of the classifier was more robust to successive removal of training data (Fig. 5b), suggesting that, in principle, fewer data would be necessary to achieve a proficient model if CC signatures are used. Of note, some molecules had a high prediction score with the GSig-based model but were nonetheless predicted to be inactive by the MFp-based counterpart, and vice versa (Fig. 5c), thus pointing to the complementarity between the SigAR and SAR approaches. Indeed, CC chemistry levels were not among the best explanatory signature types for the SR-HSE classifier. Instead, HTS bioassays (B5) and cell morphology data (D4) appeared to be more informative (Fig. 5d), an observation that is also apparent when active molecules are laid out on the B5 and D4 2D maps (Fig. 5e).

Figure 5f demonstrates that GSigs are generally favorable to MFps across the 12 toxicity pathways defined in the Tox21 benchmark data set, with particularly large differences for the SR-p53, NR-Aromatase, NR-AR, NR-PPAR-gamma, and SR-HSE tasks, and essentially the same performance for the NR-AhR and NR-ER tasks. Supplementary Figs. 16–20 give further details for these classifiers, supporting the robustness of the SigAR approach

and demonstrating that, depending on the classification task, the model will benefit from specific CC signature types (Fig. 5e and Supplementary Figs. 19, 20). The NR-AhR model, for instance, mostly leverages the chemical levels (A), whereas SR-ATAD5 benefits from cell sensitivity data (D2), and NR-ER-LBD exploits the functional (e.g., biological process (C3)) information contained within the network levels of the CC.

More comprehensively, in Fig. 5g we evaluate the predictive power of the SigAR classifiers across the full collection of MoleculeNet benchmark data sets, comprising 806 prediction tasks (Supplementary Table 1). Our SigAR predictions were generally more accurate than the equivalent chemistry-based models, meaning that our signaturizers feed additional, valuable information to a broad range of activity-prediction tasks. We observed a remarkable added value of the SigAR methodology for the physiology benchmark data sets (e.g., SIDER and ClinTox), which are, a priori, those that should benefit most from an integrative (data-driven) approach like ours. Overall, we observed 8.5% median improvements in performance with respect to chemistry-based classifiers (IQR: 1.4–19.5%, Wilcoxon's test $P$-value $= 5 \cdot 10^{-60}$) (Fig. 5h). This implies a median reduction of the gap between the actual and perfect (ideal) performance of 17.6% (IQR: 24.4–31.5%). Reassuringly, considering only molecules with reported bioactivity (i.e., included in the CC) further accentuated the difference in performance (Supplementary Fig. 21), highlighting the importance of data integration methodologies to overcome the limitations of a classical (chemistry-only) approach. Finally, it is worth noting that the superior performance of CC signatures is robustly maintained when benchmarked against different chemical descriptors or classifiers (Supplementary Fig. 22).

## Discussion

Drug discovery is a funneling pipeline that ends with a drug being selected from a starting pool of hundreds of thousands, if not millions, of compounds. Computational drug discovery (CDD) methods can aid in many steps of this costly process[29], including target deconvolution, hit-to-lead optimization, and anticipation of toxicity events. An efficient mathematical representation of the molecules is key to all CDD methods, 2D structural fingerprints being the default choice in many cases.

The renaissance of (deep) neural networks has fueled the development of novel structure featurizers[30] based on graph/ image convolutions of molecules[31–33], the apprehension of the SMILES syntax[34], or even a unified representation of protein targets[35]. These techniques are able to identify problem-specific patterns and, in general, they outperform conventional chemical fingerprints. However, neural networks remain challenging to deal with, and initiatives such as DeepChem are contributing to making them accessible to the broad CDD community[36]. The CC approach presented here shares with these initiatives the will to democratize the use of advanced molecular representations. Our approach is complementary in that it does not focus on optimally encoding chemical structures. Instead, we have undertaken the task of gathering, harmonizing, and finally vectorizing the bioactivity data available for the molecules in order to embed a wide array of bioactivities in a compact descriptor.

Since CC signatures are simple 128D-vectors, they are compatible with other CDD toolkits that primarily use multi-dimensional descriptors to represent molecular structures. This compatibility presents a unique opportunity to inject biological information into similarity searches, visualization of chemical spaces, and clustering and property prediction, among other widely used CDD tasks.

In this study, we showed how CC signatures can be used to navigate the chemical space in a biological-relevant manner, revealing somehow unexpected high-order structure in poorly

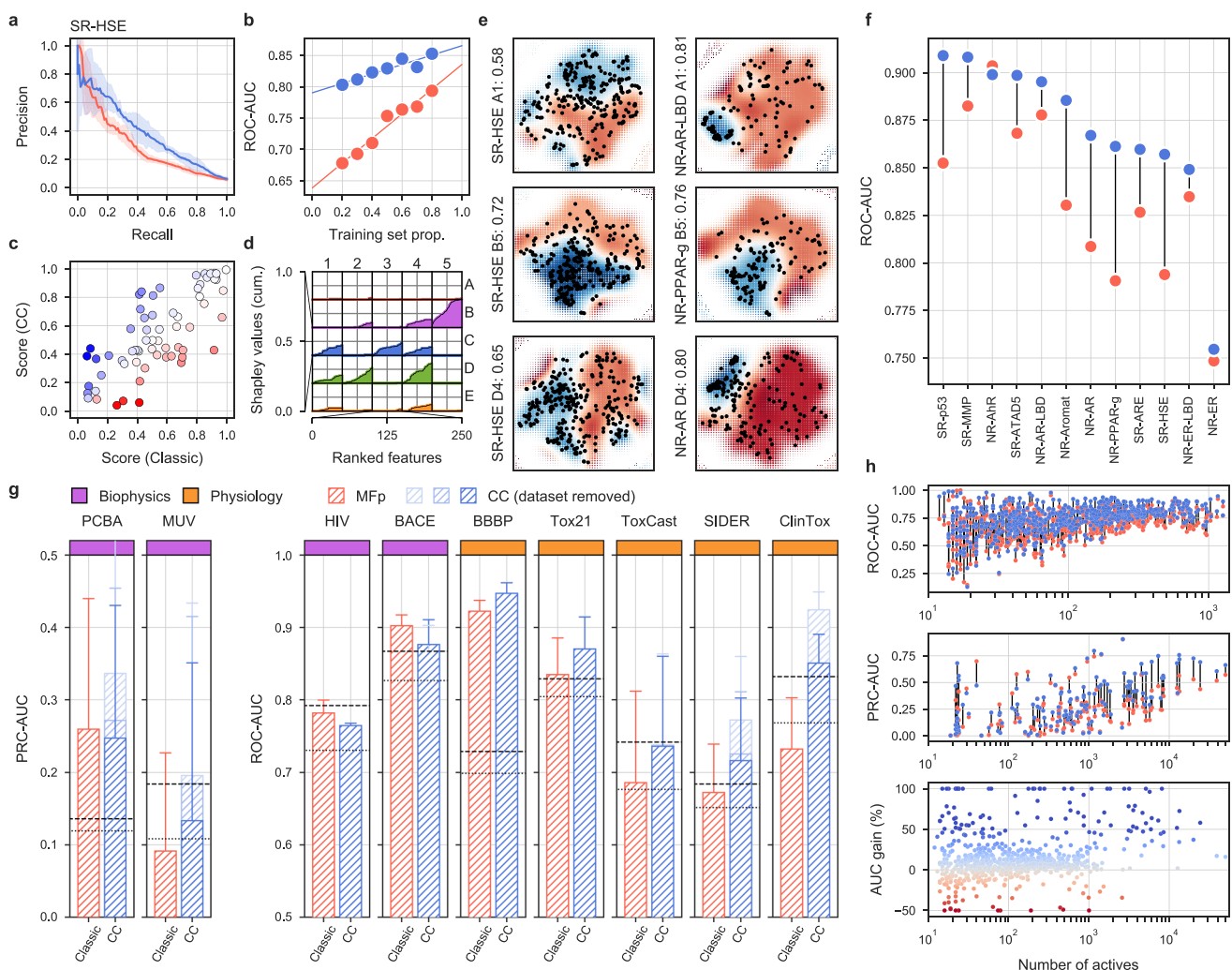

**Fig. 5 MoleculeNet benchmarks, comparing the predictive power of CC signatures with a classical MFp-based approach. a** Precision–recall curves (PRCs) for the Tox21 SR-HSE task, trained with CC signatures (blue) and MFps (red). Shaded areas span the standard deviation over five stratified train-test splits, the darker lines indicate the mean value. **b** Robustness of the SR-HSE classifier, understood as the maintenance of performance (ROC-AUC) as fewer training samples become available. **c** Prediction scores (probabilities) of active test molecules using MFps ($x$ axis) or CC signatures ($y$ axis). **d** Importance of CC data sets for the predictions. Features are ranked by their absolute Shapley value (SHAP) across samples (plots are capped at the top 250 features). For each CC data set ($S_i$), SHAPs are cumulatively summed ($y$ axis; normalized by the maximum cumulative sum observed across CC data sets). **e** 2D projections related to SR-HSE (first column) and other (second column) tasks, done for the A1, B5, and D4 CC categories (rows). A simple support vector classifier (SVC) is trained with the ($x,y$)-coordinates as features in order to determine an activity-decision function. Performance is given as a ROC-AUC on the side of the plots. Blue and red areas correspond to likely active and likely inactive regions, respectively. Active compounds are overlaid as black dots. **f** Performance of CC signatures (blue) and MFps (red) on the 12 Tox21 tasks. Tasks are ranked by their CC ROC-AUC performance. **g** Global performances of biophysics (purple) and physiology (orange) benchmark tasks. PRC and ROC AUCs are used, following MoleculeNet recommendations, the number of tasks of each category varies and is reported in the original MoleculeNet report. Here we report mean ± SD. Shades of blue indicate whether all 25 CC data sets were used (light) or whether conservative data set removal was applied (darker) (Supplementary Table 1). Dashed and dotted lines mark respectively the best and average reported performance in the seminal MoleculeNet study[13]. **h** Relative performance of CC and MFp classifiers across all MoleculeNet tasks (split by ROC-AUC and PRC-AUC metrics, correspondingly; top and middle panels). Higher performances are achieved when more active molecules are available for training ($x$ axis). The average gain in AUC is plotted in the bottom panel.

annotated natural product collections. We also demonstrated that inferred bioactivity signatures are useful to annotate mostly uncharacterized chemical libraries and enrich compound collections for activity against a drug-orphan target, beyond chemical similarities. Moreover, compared to using chemical information alone, we observed a superior performance of SigAR models across a series of biophysics and physiology activity-prediction benchmark data sets. We chose to train models with minimal parameter tuning, illustrating how our signaturizers can be used in practice with minimal knowledge of machine-learning to obtain state-of-the-art performances.

A limitation of CC signatures is that they are difficult to interpret in detail. That is, the underlying data points (binding to receptor $x$, occurrence of drug side effect $y$, etc.) cannot be deconvoluted from the 128D signature. This caveat is common to other machine-learning applications (e.g., natural language processing) where embedded representations of entities are favored over sparser, more explicit ones[37]. Nonetheless, we show that CC signatures can be interpreted at a coarser level, indicating which signature types are more informative for a certain prediction task. Another caveat of our approach is the likely existence of null signatures corresponding to innocuous molecules with no actual

bioactivity in a given CC data type[38]. Likewise, the accuracy of the signatures may vary depending on the molecule. To control for these factors, CC signatures are accompanied by an applicability score that estimates the signature quality on the basis of the amount of experimental data available for the molecule, the robustness of the prediction, and the resemblance of the predicted signature to signatures available from the training set.

Contrary to most chemical descriptors, CC signatures evolve with time as bioactivity measurements accumulate in the databases. We will release updated versions of the signaturizers once a year and, as developers of the CC, we are committed to keeping abreast of the latest phenotypic screening technologies and chemogenomics data sets. Although the current version of the CC is constrained to 25 categories, our resource is prepared to accommodate new data types, offering the opportunity to customize and extend the current repertoire of signaturizers. The growth of the CC resource is restricted by the number and quality of publicly accessible data sets, a limitation that is likely to be ameliorated with the implementation of private-public partnerships and the general awareness that, in the markedly gene-centric omics era, the depth of small-molecule annotation lags behind genomes and proteomes[39,40]. The ever-growing nature of chemical matter (in contrast to the finite number of genes) demands computational methods to provide a first estimate of the biological properties of compounds[41]. We believe that CC signaturizers can bridge this gap and become a reference tool to scrutinize the expected bioactivity spectrum of compounds.

## Methods

**Data collection**. Experimental CC signatures were obtained from the CC repository (version 2019/05). Drug Repurposing Hub molecules and annotations were downloaded from https://clue.io/repurposing (June 2019). HMDB and FooDB data were downloaded from http://hmdb.ca and http://foodb.ca, respectively (April 2020). Plant ingredients were collected from CMAUP (July 2019) and cross-species metabolites from https://www.ebi.ac.uk/metabolights (April 2020). MoleculeNet benchmark data sets were downloaded from http://moleculenet.ai in June 2019. The remaining compound collections were fetched from ZINC catalogs (http://zinc.docking.org) (June 2020).

**Siamese neural networks**. We carried out all procedures specified below for each CC data set ($S_i$) independently, and we trained 25 SNNs based on existing CC signatures and molecule triplets reflecting $S_i$ similarities. SNNs use the same weights and neural architecture for the three input samples to produce comparable output vectors in the embedding space.

Covariates matrixWe trained a SNN having horizontally concatenated signatures ($S_1$–$S_{25}$) as a covariates matrix ($X$), and producing 128D-vectors as output ($Y$). The covariates matrix was stacked with a pre-compressed version of CC signatures (named signatures type II) with 128 dimensions. Only CC data sets covering at least 10% of $S_i$ were stacked in $X$. Thus, given $n$ molecules in $S_i$, and having $m$ $S_{1-25}$ data sets cross-covering at least 10% of $n$, $X$ would be of shape ($n$, $128 \cdot m$). For each molecule (row), missing signatures were represented as not-a-number (NaN) values.

*Triplet sampling*. We sampled $10^7$ molecule triplets (i.e., $10^7/n$ triplets per anchor molecule). Positive samples (i.e., molecules similar to the anchor) were drawn using the FAISS k-nearest neighbor search tool[42]. The value of $k$ was empirically determined so that it maximized the average ROC-AUC of similarity measures performed against the rest of CC data sets, and it was then clipped between 10 and 50. Negative samples were randomly chosen from the pool of molecules at larger distances than the positive compounds.

*SNN architecture*. SNNs were built and trained using Keras (https://github.com/fchollet/keras). After the $128 \cdot m$-dimensional input layer, we added a Gaussian dropout layer ($\sigma = 0.1$). We then sequentially added two fully connected (dense) layers whose size was determined by the m magnitude. When $m \cdot 128$ was higher than 512, the two hidden layers had sizes of 512 and 256, respectively. For smaller m values, we linearly interpolated the size between input and output (128) dimensions (e.g., for $m = 7$, the two hidden layers had sizes of 448 and 224, respectively). Finally, a dense output layer of 128 dimensions was sequentially added. For the hidden layers, we used a SeLU activation with alpha-dropout regularization (0.2), and the last (output) layer was activated with a Tanh function, together with an L2-normalization.

*Signature dropout*. We devised a dropout strategy to simulate the availability of CC signatures at prediction time. To do so, we measured the proportion of experimental $S_{1-25}$ signatures available for not-in-$S_i$ molecules. These observed (realistic) probabilities were then used to mask input data at a fitting time, more

frequently setting those CC categories with the smaller probabilities to NaN. The $S_i$ signature was dropped out with an oscillating probability (0–1) over the training iterations (5000 oscillation cycles per epoch).

*Loss functions*. To optimize the SNN, we used a pair of loss functions with a global orthogonal regularization[43]. The first one was a conventional triplet loss, checking that the distance between the anchor and the positive measured in the embedding (128D) space was shorter than the anchor-negative distance (margin = 1). The second loss was exclusively applied to the anchor molecule, and it controlled that the embedding resulting from the signature dropout was similar to the embedding obtained using $S_i$ alone (mean-squared error (MSE)). Global orthogonal regularization (alpha = 1) was used to favor the maximal spread-out of signatures in the embedding space. The Adam optimizer was used with a default learning rate of $10^{-4}$.

*Evaluation*. For each $S_i$, we split the list of $n$ molecules into the train (80%) and test (20%) sets. Splitting was done after removing near-duplicates with FAISS. We then defined three triplet splits, i.e., train–train, test–train, and test–test, using molecules from the train and test sets as anchors and positives/negatives, correspondingly. For CC spaces with <30,000 molecules, we trained the model for 5 epochs, whereas the largest data sets were trained for 2 epochs. Two accuracy measures were defined: (a) a triplet-based accuracy quantifying the proportion of correctly classified triplets by Euclidean distance measurements in the embedding space (dropping out $S_i$); and (b) an anchor-based accuracy measuring the correlation between the $S_i$-dropped-out embedding and the $S_i$-only embedding. Given the bimodal distribution endowed by the Tanh activation, we chose to use a Matthews correlation coefficient (MCC) on a contingency table of binarized data (positive/negative along the 128 dimensions).

*Light-weight signaturizers*. We ran predictions for all molecules available in the CC universe ($N = 778,531$), producing 25 matrices of shape ($N$, 128). These matrices were used to learn chemistry-to-signature (CTS) signaturizers that are easy to distribute, allowing us to obtain signatures for a given molecule on-the-fly. CTS signaturizers were trained on a large number of molecules ($N$) with the aim to approximate the pre-calculated signatures presented in this work. Thus, in practice, a CTS signaturizer will often act as a mapping function, since the number of pre-calculated signatures is very large and covers a considerable portion of the medicinal chemistry space. CTS signaturizers were trained for 30 epochs and validated with an 80:20 train–test split, using 2048-bit Morgan Fingerprints (radius = 2) as feature vectors. Three dense hidden layers were used (1024, 512, and 256 dimensions) with ReLU activations and dropout regularization (0.2). The output was a dense layer of 128 dimensions (Tanh activation). The Adam optimizer was used (learning rate = $10^{-3}$). CTS signaturizers achieved a correlation with the type III signature of 0.769 ± 0.074.

**Applicability domain estimation**. An applicability score ($\alpha$) for the signatures can be obtained at prediction time by means of a linear combination of five factors related to three characteristics that help increase trust in the predictions. These factors were tuned and calibrated on the test set.

*Distance*. Signatures that are close to training-set signatures are, in principle, closer to the applicability domain. We measured this distance in an unsupervised way (i.e., the average distance to 5/25 nearest-neighbors) and in a supervised way by means of a random forest regressor trained on signatures as features and prediction accuracy (correlation) as a dependent variable. In addition, we devised a measure of intensity, defined as the mean absolute deviation of the signatures to the average (null) signature observed in the training set.

*Robustness*. The signature-dropout procedure presented above can be applied at prediction time to obtain an estimate of the robustness of the prediction. For each molecule, we generated 10 dropped-out inputs, thereby obtaining an ensemble of predictions. Small standard deviations over these predictions indicate a robust output.

*Expectancy a priori*. We calculated the accuracy that is expected given the input signatures available for a particular molecule. Some CC signature types are highly predictive for others; thus, having these informative signatures at hand will in principle favor reliable predictions. This prior expectancy was calculated by fitting a random forest classifier with 25 absence/presence features as covariates and prediction accuracy as an outcome.

**Validation of the inferred bioactivity signatures and the signaturizers**. To further explore the validity of the inferred bioactivity signatures and the developed signaturizers, we ran three independent technical validations.

First, we conducted a scrambling experiment that involved the calculation of GSigs related to randomized MFps, in order to assess whether the resulting GSigs would have some sort of structure (signal) or not. This is, we randomly picked 1000 compounds from the CC universe and computed their GSigs with and without a scrambling of their structural representation (i.e., their Morgan fingerprints). Then, we calculated all the pairwise distances signatures in each set. As expected, when we plot the shortest distance between pairs of molecules within the GSig and scrambled-GSig sets, we see that there are no significant similarities between

scrambled and global signatures, while GSigs can indeed detect similarities between small molecules, i.e., short distances between GSigs (Supplementary Fig. 23a). Additionally, 2D projections (t-SNE and PCA) of scrambled and global signatures (Supplementary Fig. 23b) reinforce the observation that scrambled signatures are indeed different from real GSigs and contain no bioactivity signal.

Second, we performed a Y-scrambling experiment on the whole array of MoleculeNet prediction tasks (previously introduced). Y-scrambling (or Y-randomization) is commonly used to validate QSPR/QSAR models by training predictors on the randomly shuffled dependent variable (Y) that we aim to predict. Comparing the performances (on an unshuffled test set) of models trained on shuffled vs unshuffled data we can assess the robustness of descriptors and rule out the effect of random correlations. This is exactly what we observe for both GSig and ECFP4 descriptors (Supplementary Fig. 24), with a drop of model performances to an average of 0.5 ROC-AUC when training on scrambled data.

Lastly, we ran a time-series experiment by confronting the small-molecule bioactivity signatures predicted using the 2019_01 (September 2019) signaturizers with novel experimental bioactivity data available in the newly released 2020_02 (November 2020) version. In this new CC release, most of the source databases have been updated presenting new data for molecules that were not present in the previous versions.

Profiting from the novel experimental data, for each bioactivity space, we gathered molecules only present in the new version and that are completely novel in terms of CC annotation. Using the signaturizer 2019_01, we predicted signatures for the new molecules and searched for neighbors in the 2019_01 CC universe. We then compared this set of neighbors to those confirmed in the 2020_02 version, excluding molecules not available in the 2019_01 CC universe (i.e., those neighbors that are also new molecules). Supplementary Fig. 25 shows the fraction of novel molecules (y axis) for which at least one correct neighbor is identified among the top [1–1000] predicted neighbors (x axis), within the roughly 1 M molecules in the CC chemical space. We also show the same fraction of recovered real neighbors when randomizing the signaturized molecules. Despite the limited number of new molecules with experimental information in some of the spaces, we observe that the signaturizers derived from previous versions can identify similar molecules for a significant fraction of the new compounds. Moreover, we also see how increasing the applicability score threshold augments the reliability of the predicted signatures in all the bioactivity spaces. However, as expected, when we use random molecules as bait, we cannot identify true neighbors and there is no relationship with the applicability scores.

## Library enrichment for activity against Snail1

*Computational screening. Compound collections*: Two compound collections were considered for screening, namely the IRB Barcelona library (17,563 compounds, considering the connectivity layer of the InChIKey) and the commercial Prestwick library (1108 compounds). Of these, 627 and 1104 were part of the CC universe, respectively, meaning that they had some type of reported bioactivity.

*Chemical query*: This query involved the search for compounds that were chemically similar to curated DUB inhibitors, based on their known activity on promising DUBs according to a previous siRNA/Snail1 screen[20] (Supplementary Data 1), or similar to DUB inhibitors belonging to a broader list (with DUB-binding data available). The query was implemented by computing chemical similarity (best across A1 + A4, P < 0.001) to DUB inhibitors from the literature (curation categories 1 and 2 in Supplementary Data 1, corresponding to 6 DUB inhibitors). In total, this query selected 169 compounds.

*Biological queries*: In addition to DUBs, we considered other proteins relevant to Snail1 activity, namely TGFBR1/2, ERK2, FBXL5/14, DDR2, and GSK3B (Supplementary Data 1). We then looked for transcriptional signatures associated with the corresponding genes in the L1000 Connectivity Map (shRNA assays, reversed over-expression assays, and known small-molecule perturbagens) and also in CREEDS, which brings together data from GEO[44]. Overall, we gathered 132 transcriptional signatures with a potential of having a connection to Snail1 (Supplementary Data 1). Different priorities (0–4) were given to these signatures based on our mechanistic knowledge of Snail1 (Supplementary Data 1 legend). Transcriptional signatures were converted to the CC D1 format as explained above. In addition, we derived C3–5 signatures for the Snail1-related genes, including DUBs highlighted by the siRNA/Snail1 screen.

We looked for connectivities (similarities, P-value < 0.001) between signatures of compounds in the D1 space and the list of Snai1-related signatures (at least 10 up/downregulated genes per signature). We did two searches (search H and search L), one against high-priority signatures (priority ≥3), and another with a more relaxed cutoff (priority ≥1). In parallel, we derived C3–5 signatures for non-DUB Snail1-related proteins (TGFBR1/2, etc.).

*Random query*: Molecules were randomly picked from the PWCK and IRB libraries, proportionally to the relative abundance of molecules from the two libraries in the lists retrieved from the previous queries (Supplementary Data 1).

*Comparison to classical ECPF4-based queries*: To compare our results to a more classical compound selection strategy, based on ECPF4 similarities, we calculated all Tanimoto similarities between the >17k compounds in our library and the 45 annotated DUB inhibitors, and selected the 222 compounds with the highest similarities. Additionally, for each compound, we also computed the average

similarity to the closest 5 DUB inhibitors and, again, selected the 222 with the highest scores. The Supplementary Fig. 26a shows the overlap between the different selections, revealing that all three approaches are indeed complementary. We then calculated the enrichment in the identification of Snail1 inhibitors achieved by each strategy considering as positive hits all those compounds able to decrease 1.5 times the levels of Snail1, and compared the hits found by each strategy to those found in random compounds. As Supplementary Fig. 26b shows, the signatures-based approach achieved a ~3.5-fold enrichment, closely followed by the average score of compounds over the 5 closest DUB inhibitors (~2.8-fold) and the one that only considers the highest Tanimoto similarity with any DUB inhibitor (~1.7-fold). Overall, the three approaches show a significant capacity to enrich compound libraries for a given function but, as expected, the combination of chemical and biological signatures identifies additional compounds.

*Cells*. We used MDA-MB-231 cells stably transduced with pLEX-Snail1-Firefly Luciferase and pMSCV-Renilla Luciferase from our previous study[20], and cultured them in DMEM supplemented with 10% FBS, glutamine, and antibiotics (ThermoFisher Scientific).

*Dual-luciferase assay screening*. We seeded $5·10^4$ cells in 96-well white plates prepared for cell culture (Corning). The day after, pre-diluted compounds of the chemical libraries were added to the cells at a final concentration of 20 μM, or in a few cases, of 4 μM, depending on the stock concentration and the maximum amount of DMSO that could be used in the assay. Several replicas of the vehicle controls (DMSO) or the positive control (the general DUB inhibitor PR-619 (Sigma-Aldrich)) were distributed along the experimental plates to allow internal normalization. After 6 h of incubation, the medium was removed. Cells were then directly lysed with passive lysis buffer (Promega), and plates were stored at −20 °C. Firefly and Renilla luciferase were quantified using the Dual-Luciferase Reporter assay system (Promega) in a GloMax luciferase plate reader (Promega). Four replicas conducted on two days were performed.

Intensities were corrected for each measurement (i.e., Firefly and Renilla) using one linear model per replica. The linear model included plate, row, and column (as ordinal covariates) and type of measure (namely compounds, negative and positive controls) as fixed effects, as well as plate-row and plate-column interactions. Estimation of effects for plate, row, and column (and their interactions) were used to correct intensity values. Intensities were previously transformed (square root) in order to fulfill the assumptions of linear models. In practice, this transformation implies a correction based on the median (instead of mean) effects, and it is thus robust to outliers (potential hits). Corrected values were transformed back to the original scale of the measures after correction. For normalization against controls, log2-ratios of intensities were computed against the mean of negative controls within each marker-replicate. Log2-ratios of Firefly:Renilla were then computed for signal evaluation.

The enrichment of hit rates was evaluated separately for each query (chemical, biological) with respect to the random distribution of Firefly:Renilla ratios.

To ensure that the tested compounds did not directly interfere with luciferase activity, we devised a double computational and experimental strategy. On the one hand, we used the available PubChem bioassay AID:411 (https://pubchem.ncbi.nlm.nih.gov/bioassay/411) listing inhibitors of Firefly Luciferase to train a simple predictor of luciferase inhibition. The Bioassay contains over 70 thousand inactive and over 1500 active compounds against luciferase for which we derived both the GSig (presented in our manuscript) and classical Morgan Fingerprint (ECFP4 descriptor). A simple logistic regression (with class weights to handle active/inactive imbalance) was sufficient to achieve good classification accuracy. We performed a 5-fold stratified cross-validation and we measured performance in terms of ROC-AUC and F1-score. We report the result of both train and test splits in Supplementary Fig. 27a. We then used the classifier trained on GSigs to calculate the probability of luciferase interaction for the 400 tested compounds, including the top 25 candidates we identified. Supplementary Fig. 27b shows that an interaction of these compounds to the luciferase reporter is unlikely. Unfortunately, we did not find training data of similar characteristics for Renilla luciferase; no PubChem bioassay provides inhibition data specifically for this enzyme. However, please note that in our experimental measure of activity (i.e., low Firefly:Renilla ratio), false positives may reveal Firefly luciferase inhibition (numerator), not Renilla luciferase inhibition (denominator).

On the other hand, we used transiently transfected MDA-MB-231 cells expressing constitutively the Firefly and the Renilla Luciferases under the control of CMV and TK promoters, respectively. Specifically, MDA-MB-231 cell line was cultivated in DMEM-F12 medium. Cells were co-transfected with 2 μg of pCMV-luc and 6 μg of pTK-RN in 100 mm plates, using Polyethylenimine (PEI) at a ratio DNA:PEI 1:3. 24 h after transfection, cells were trypsinized and seeded in a 96-well plate at 40,000 cells per well. 48 h after transfection, cells were treated with the drugs at 20 μM during 6 h. Cells were then lysed using the Passive Lysis Buffer provided in the Dual-Luciferase Reporter Assay System (Promega) and Firefly and Renilla activities were measured following the manufacturer's instructions. As can be seen in Supplementary Fig. 27c, only the positive control (PR619) at a high concentration showed a moderate interference with the Renilla luciferase activity. Indeed, none of the selected compounds showed an inhibitory effect neither on Firefly nor on Renilla luciferase activity, confirming that the measured signal was not confused by interference with the reporter enzymes.

**Signature–activity relationship (SigAR) models**. For each classification task in the MoleculeNet, we sought to predict active/inactive (1/0) compounds using horizontally stacked CC signatures. A random forest classifier was trained using hyperparameters identified with HyperOpt[45] over 10 iterations (number of estimators: (100, 500, 1000), max depth: (None, 5, 10), minimum sample split: (2, 3, 10), criterion: (gini, entropy), maximum features: (square root, log2)). Classifiers were calibrated using a Mondrian cross-conformal prediction scheme over 10 stratified splits. The evaluation was done with five stratified 80:20 train–test splits. Large MoleculeNet data sets such as PCBA were trained on a maximum of 30 under-sampled data sets, each comprising 10,000 samples. Scaffold-aware stratified splits, when necessary, were done ensuring that Murcko scaffolds[46] observed in the training set were not present in the test set[47]. Please, note that we followed MoleculeNet recommendations strictly, both in terms of splitting (e.g., scaffold-based) and in terms of performance measure (AUROC/AUPR) to ensure a fair assessment.

Signature importance for each prediction was calculated by aggregating Shapley values (SHAP) as follows. First, features were ranked by their absolute SHAP across molecules. We then calculated the cumulative rank specific to each signature type ($S_i$) (up to 250 features). Signature types with more of their dimensions in highly ranked positions were deemed to be more explanatory for the prediction task.

To assess the robustness of our results, we extended the collection of chemical descriptors beyond ECFPs. In particular, we included Daylight-like (RDKit) fingerprints (path-based), MACCS keys, and a data-driven state-of-the-art descriptor named CDDD, which is based on a deep-learning model trained on string (SMILES) representations of the molecules. Additionally, we repeated the SigAR task predictions with a model-agnostic approach based on the AutoML TPOT methodology. In brief, TPOT automatically performs feature selection/processing, classifier choice, and hyperparameter optimization across a wide array of standard ML techniques. Thus, this approach provides a fair (ML-independent) head-to-head comparison between our descriptors and the rest of the chemical fingerprints. Note that, in this case, we could not address the PCBA MoleculeNet subtasks (involving 400k molecules) due to prohibitive computational costs for TPOT.

**Reporting summary**. Further information on research design is available in the Nature Research Reporting Summary linked to this article.

## Data availability

The latest version of inferred signatures (version 2020_02 at the moment of writing this manuscript) is available for direct download from https://chemicalchecker.com/downloads. Additional data that support the findings of this study are available from the corresponding author upon reasonable request.

## Code availability

Software for generating CC signatures is available as a python package at http://gitlabsbnb.irbbarcelona.org/packages/signaturizer. The signaturizer API allows the conversion of molecules (represented as SMILES or InChI strings) to the 25 signature types available from the CC. These pre-trained signaturizers are light-weight versions of the SNNs presented here, freeing the user from the need of setting up a full version of the CC (see "Methods" section). Signaturizers are available as TensorFlow Hub SavedModel instances and are automatically downloaded by the API the first time they are used. The full CC repository is open-sourced at http://gitlabsbnb.irbbarcelona.org/packages/chemical_checker and also available on Zenodo[48].

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

## Acknowledgements

We would like to thank the SB&NB lab members for their support and helpful discussions. We are grateful to T.O. Botelho, I. Ramos, and C. Gonzalez for giving us access to the IRB Barcelona and Prestwick libraries. P.A. acknowledges the support of the Generalitat de Catalunya (RIS3CAT Emergents CECH: 001-P-001682 and VEIS: 001-P-001647), the Spanish Ministerio de Economía y Competitividad (BIO2016-77038-R), the European Research Council (SysPharmAD: 614944), and the European Commission (RiPCoN: 101003633). A.G.d.H. acknowledges support by Agencia Estatal de Investigación (AEI) and Fondos FEDER (PID2019-104698RB-I00).

## Author contributions

M.D.-F. and P.A. designed the study and wrote the manuscript. M.B., M.D.-F., P.B.-i.-M., M.O.-R., and O.G.-P. implemented the entire computational strategy. E.P., V.A., V.M.D., A.B.-L.l., I.B.-H., N.V., and A.G.d.H. performed and analyzed the Snail1-luciferase assays. All authors analyzed the results and read and approved the manuscript.

## Competing interests

The authors declare no competing interests.
