## [Peer Review File · Nature Communications]

Reviewers' Comments:

Reviewer #1:

Remarks to the Author:

The paper by Bertoni et al. is an extension of a recent paper by the same group (ref. 6) describing the chemical checker (CC), which is a database of bioactive compounds grouping chemical and biological information. In this new paper the authors used a neural network to fill the gaps in the CC data, i.e. to infer missing biological data and add it to CC. To make the point that this approach is valuable, the authors describe the identification of deubiquitinase (DUB) inhibitors that indirectly downregulate Snail1 in cells, as revealed by a luciferase reporter system. The paper is very well written and richly illustrated with figures, and some supporting data is provided. Although the work is very close to their ref. 6, the idea to augment the CC data using a neural network is new. The work is interesting and might become publishable if the following issues can be addressed, which mostly concern the validation experiment presented as demonstration that the approach is useful:

1. Describing the cpds as Snail1 targeting is a bit misleading: in truth, the authors just searched for close analogs of DUB inhibitors, of which there are already many described examples.
2. After a screening pipeline the authors focus on 25 compounds. The structure of these 25 hits must be shown, for the moment only four structures are shown and the others are not present anywhere.
3. Luciferase-based assays can reveal luciferase inhibitors. Therefore, an independent, second assay not depending on luciferase must be reported to be sure that the hits are true DUB inhibitors.
4. The reported compounds 6, 7, 8, and 9 share large substructures with the 45 DUB inhibitors. For example, 7 is almost identical to Mitoxantrone, and 8 and 9 share the same scaffold with Pimozide, where Mitoxantrone and Pimozide are both in the list of know DUB inhibitors. This should be stated, and the nearest neighbour among the DUB known analogues should be reported.
5. Dimensionality reduction of the ECFP to 128 dimensions is too much compression, this does not exploit the full potential of chemical fingerprints. This must be discussed.
6. Only three compounds are found exclusively with bioactivity queries, so it is fundamental to better discuss them. Which are their structures (see also comment 2)? Do they differ a lot from the known DUB inhibitors? Are they large compounds? If yes, a comparison should be carried out with structural fingerprints suitable for large molecules to see if they might be found.
7. As control please report how many of the top compounds for Snail1 are found with a "classical virtual screening" of the PWCK and IRB libraries using the 45 known DUB compounds as queries and ECFP4 as standard structural fingerprint.
8. In the paragraph "Enhanced prediction capabilities compared to chemical descriptors" a model (SigAR) is compared with a set of fingerprints (ECFPs). Therefore, one cannot conclude that the better performance is due to the use of bioactivity fingerprints. To establish so, one should use the bioactivity signature as fingerprint within the models of the benchmark (or in general compare the two types of fingerprints with the same model). Furthermore, ECFPs are not the only option in terms of "chemistry fingerprints", and they are not the only descriptors available in MoleculNet, the authors should extend their benchmark to (at least) all available fingerprints in MoleculNet to demonstrate that bioactivity fingerprints perform better or different than established fingerprints.
9. Extending the description of the bioactivity fingerprint would increase the readability of the manuscript.
10. In figure 2g how is the chemical diversity assessed? It is unclear what low and high refer to.

11. Figure 2h, legend. Is the cluster enriched with heat shock protein 90 (HSP90AA1) ...” I assume a word is missing, inhibitors perhaps? And molecules 4 and 5 are a good example of similarity/dissimilarity to which compound/ set of compounds?

12. Figure 3. How does the chemical space look like when using structural fingerprints? Do compounds still cluster by target? In these cases, is the awareness of the target coming from the bioactivity fingerprint or is it intrinsic in the chemical structure?

Reviewer #2:

Remarks to the Author:

Bertoni et al describe a new approach for the prediction of bioactivity for small organic compounds. While existing methods in the field mostly focus on chemical descriptor like fingerprints, the authors try to capture all different types of data from chemical structure to known bioactivity data. Since data is sparse for many descriptor types, they use DNNs to predict missing values. For validation, the authors compare their approach named Chemical Checker (CC) to a series of prediction tasks defined in DeepChem. Furthermore, they apply it Snail1 and report new chemical matter found.

In general, machine learning for bioactivity prediction is a very hot topic which is not surprising given the advances in ML and the relevance of the prediction task for drug discovery. The study presented has - to the best of my knowledge - a new element, namely the integration of heterogeneous data into one system. It has been shown before, that transfer learning is helpful and maybe one of the key strength of DNNs in cases of sparse data like we have in early drug discovery. Therefore, the manuscript content deserves publication although the gain in prediction quality is in my view more incremental than ground breaking. I recommend the authors to address several issues before publication:

- In the introduction, the authors mostly ignore the whole field of chemogenomics. There are numerous approaches based on the idea of predicting bioactivity from bioactivity data. The first, to my knowledge, is from Kauvar in 1995 (Chem.Biol.) which used bioactivity data as fingerprints. Mostly known is certainly the SEA approach (Keiser et al, Nature 2009). Of course, these papers have different focus, but they are of importance to put this work into the right scientific context.

- The whole structure of the manuscript makes it very hard to follow. Almost all details to really understand the methods are not where they are needed. The manuscript would greatly benefit from a clear method description including important parametrization. Since the authors publish source code, this might not hinder reproducibility, but it certainly hinders to get a good understanding of the many technical elements of this work.

- The technical validation based on retrospective data needs substantial improvement. Complex machine learning methods like the presented one should be validated more critically, for example by γ -scrambling experiments, by showing the performance delta by skipping individual steps of the method, by cluster-cross validation or time-series analysis.

- The comparison to DeepChem predictions is technically not sound. I have never seen that descriptors are taken out instead of similar data points. For comparison, all molecules similar to the testset compounds have to be removed from all datasets used for training, otherwise the experiment is inconclusive. Furthermore, for comparison the best performing chemical descriptors which are graph descriptors rather than simple morgan fingerprints have to be taken into account.

- For the Snail1 experiment, the presented data is insufficient. The data tables contain InChI Keys rather than SMILES or another readable format making it extremely difficult to look at the molecules. There is no PAINS filter or frequent hitter analysis. There is no analysis of the similarity of the new compounds to known Snail1 inhibitors, no comparison of bioactivity levels. Furthermore, specialized journals like JMedChem, etc. do require dose response curves for all active compounds.

Point-by-point responses to Reviewer's comments

NCOMMS-20-38745-T "Bioactivity descriptors for uncharacterized compounds" by Bertoni*, Duran-Frigola*, Badia-i-Mompel*, et al.

We would like to thank the Editor and Reviewers for their critical reading of our manuscript and the positive feedback. We have now addressed in detail all the raised issues, which, we feel, has improved the quality and clarity of our paper. In brief:

- We now include in Data S1 the SMILES representation for all the compounds as well as several structural alerts (i.e. PAINS analysis) and, for the Top25 active compounds their structures as well as the structure and Tanimoto similarities to their closest known DUB inhibitor.
- We also rule out (computationally and experimentally) the possibility that any of the Top25 active compounds might interfere with luciferase activity.
- We show how chemical data represented at the different stages of our pipeline (Sign1 and the more compressed Sign2 and 3) can recapitulate the chemical similarity detected by the more explicit ECFP4.
- We provide a more detailed analysis of the specific characteristics of the compounds selected by chemical and biological signatures, which unveiled an error in their annotation in the original version of the m/s.
- We also show how a more classical (chemistry-based) virtual screening of the IRB compounds library would have selected molecules against Snail1 activity, and compare the results with our selection.
- We implement the SigAR models using a battery of different chemical descriptors and classifiers to show the overall superior performance of our bioactivity signatures.
- We also perform scrambling and time-series validations of the SigAR models, showing that they are indeed sound.
- We now show some dose-response data for the Top10 most active compounds against Snail1 activity, as well as some preliminary results of our follow-up experiments.
- Finally, we have revised our m/s to make it easier to follow, and have included some extra information to clarify the issues raised by the Reviewers.

Please find below a point-by-point response to Reviewers' comments. Major edits are highlighted in yellow in the revised version of the manuscript.

Reviewer #1:

The paper by Bertoni et al. is an extension of a recent paper by the same group (ref. 6) describing the chemical checker (CC), which is a database of bioactive compounds grouping chemical and biological information. In this new paper the authors used a neural network to fill the gaps in the CC data, i.e. to infer missing biological data and add it to CC. To make the point that this approach is valuable, the authors describe the identification of deubiquitinase (DUB) inhibitors that indirectly downregulate Snail1 in cells, as revealed by a luciferase reporter system. The paper is very well written and richly illustrated with figures, and some supporting data is provided. Although the work is very close to their ref. 6, the idea to augment the CC data using a neural network is new. The work is interesting and might become publishable if the following issues can be addressed, which mostly concern the validation experiment presented as demonstration that the approach is useful:

We would like to thank the Reviewer for assessing our m/s and, indeed, for his/her positive comments about the relevance of the work, the clarity of the presentation and for supporting its publication. Below we address his/her comments in detail.

1. Describing the cpds as Snail1 targeting is a bit misleading: in truth, the authors just searched for close analogs of DUB inhibitors, of which there are already many described examples.

The inhibition of DUBs is, indeed, a good strategy to reduce the Snail1 levels. Thus, we wanted to include this strategy in our search for Snail1 modulators, together with other types of “queries” unrelated to DUBs. In particular, as the Reviewer points out, we used known DUB inhibitors and looked for chemically similar compounds in the IRB and Prestwick libraries (i.e. *Chemical query*). As explained in the m/s, we used 45 known DUB inhibitors, 6 of which were annotated as binders of the candidate DUBs found in our previous siRNA-DUB/Snail1 assay¹. Using the chemical query, we identified compounds in our library that were, to some extent, chemically similar to known DUB inhibitors (Data S1). On the other hand, and profiting from the bioactivity signatures, we also looked for compounds that could mimic the transcriptional effects described when genetically perturbing genes/proteins known to be related to Snail1 activity (e.g. TGFBR1/2, ERK2, FBXL5/14, DDR2 and GSK3B), which included over 100 different expression experiments. We also derived networks-level signatures that captured the effects of perturbing these Snail1-related genes/proteins. Please note that these *Biological queries* are in principle unrelated to DUB inhibition. Overall, using this second strategy we retrieved over 3,000 small molecules from our compound libraries. Finally, we selected a total of 222 compounds, with 169 and 131 compounds respectively from the chemical and biological queries (with an overlap of 78 compounds) for experimental testing, together with 188 random compounds.

The Reviewer is right in his/her perception that a substantial number of the active compounds against Snail1 were indeed identified with chemical queries (i.e. similar to known DUB inhibitors). However, we would like to stress that 6 in the Top25 were also identified by the biological queries, providing a higher level of confidence and a starting point to investigate their potential mechanism of action. Perhaps more importantly, 3 compounds were only identified by the biological queries, stressing the

complementarity of the two approaches. Moreover, as illustrated in Figure 4, if we only consider those compounds picked by the biological queries, we still see a very significant enrichment with respect to the randomly selected compounds. Please, note that, even if some of these compounds are also chemically similar to DUB inhibitors, this information was not used in the biological queries that identified them.

Figure R1. The Venn diagram shows the overlap between the 169 small molecules selected from the Chemical queries, the 131 compounds selected from Biological queries and the Top25 active against Snail1, according to our dual-luciferase screen. Please, note that the Top25 active compounds also include two concentrations of the positive control (PR619) and three randomly selected compounds.

We have extended the description of the different queries and the retrieved compounds, and have added this figure as a panel in the new Figure S13.

2. After a screening pipeline the authors focus on 25 compounds. The structure of these 25 hits must be shown, for the moment only four structures are shown and the others are not present anywhere.

The Reviewers is indeed right, and we have now included a new tab in Data S1 where we show the chemical structure and SMILES representation of the Top25 compounds, together with the structure of its closest known DUB inhibitor and the Tanimoto similarity (Morgan fingerprints). Additionally, and in response to a point raised by Reviewer #2, we also included the scores of several PAINS alerts.

3. Luciferase-based assays can reveal luciferase inhibitors. Therefore, an independent, second assay not depending on luciferase must be reported to be sure that the hits are true DUB inhibitors.

This is an important issue, and we thank the Reviewer for pointing it out. To make sure that we were not identifying luciferase inhibitors, we took a double computational/experimental approach.

On the one hand, we used the available PubChem bioassay AID:411 (<https://pubchem.ncbi.nlm.nih.gov/bioassay/411>) listing inhibitors of Firefly Luciferase to train a simple predictor of luciferase inhibition. The Bioassay contains over 70 thousand inactive and over 1500 active compounds against luciferase for which we derived both the GSig (presented in our manuscript) and classical Morgan Fingerprint (ECFP4 descriptor). A simple logistic regression (with class weights to handle active/inactive imbalance) was sufficient to achieve good classification accuracy. We performed a 5-fold stratified cross-validation and we measured

performance in terms of ROC-AUC and F1-score. We report the result of both train and test splits in the figure below. Note that, in line with our findings across MoleculeNet, the GSig representation of molecules performs better in each split and scoring metric.

Figure R2. Performance of the logistic regression classifier to identify compounds with potential luciferase activity. The performance of the classifier is measured in the different train and test set splits using the area under the ROC (left) and F1 (right) metrics when using Gsig and ECFP4 molecular descriptors.

We then used the classifier trained on GSigs to calculate the probability of luciferase interaction for the 400 tested compounds, including the top 25 candidates we identified. The results reported below show that an interaction of these compounds to the luciferase reporter is unlikely (Figure R3).

Unfortunately, we did not find training data of similar characteristics for Renilla luciferase; no PubChem bioassay provides inhibition data specifically for this enzyme. However, please note that in our experimental measure of activity (i.e. low Firefly:Renilla ratio), “false positives” may reveal Firefly luciferase inhibition (numerator), not Renilla luciferase inhibition (denominator).

Figure R3. Results of the logistic regression classifier to identify luciferase activity on the set of known luciferase inhibitors (green), tested *Snail1* modulation candidates (gray) and experimentally identified actives (Top25; red). As it can be seen, according to the classifier, the probability of any of the active molecules to interfere with luciferase activity is very small.

On the other hand, we experimentally tested the effect of the Top25 compounds on luciferase activity. We used transiently transfected MDA-MB-231 cells expressing

constitutively the Firefly and the Renilla Luciferases under the control of CMV and TK promoters, respectively. Specifically, MDA-MB-231 cell line was cultivated in DMEM-F12 medium. Cells were co-transfected with 2 μ g of pCMV-luc and 6 μ g of pTK-RN in 100mm plates, using Polyethylenimine (PEI) at a ratio DNA:PEI 1:3. 24h after transfection, cells were trypsinized and seeded in a 96 well plate at 40,000 cells per well. 48h after transfection, cells were treated with the drugs at 20 μ M during 6 hours. Cells were then lysed using the Passive Lysis Buffer provided in the Dual Luciferase Reporter Assay System (Promega) and Firefly and Renilla activities were measured following the manufacturer instructions.

As can be seen in Figure R4, only the positive control (PR619) at a high concentration showed a moderate interference with the Renilla luciferase activity. Indeed, none of the selected compounds showed inhibitory effect neither on Firefly nor on Renilla luciferase activity, confirming that the measured signal was not confused by interference with the reporter enzymes.

Figure R4. Results of the effect of the Top25 compounds on MDA-MB-231 cells expressing constitutively the Firefly and the Renilla Luciferases under the control of CMV and TK promoters, respectively. The bar plots show the average results of two experiments done in triplicate.

We have included two paragraphs in the Online Methods describing our approaches to discard a direct interference of the compounds with luciferase activity and Figure S26 that summarizes the results.

4. The reported compounds 6, 7, 8, and 9 share large substructures with the 45 DUB inhibitors. For example, 7 is almost identical to Mitoxantrone, and 8 and 9 share the same scaffold with Pimozide, where Mitoxantrone and Pimozide are both in the list of known DUB inhibitors. This should be stated, and the nearest neighbour among the DUB known analogues should be reported.

Compound 7 is, indeed, mitoxantrone as we happened to have it in our compounds library and, as we report it in the text, we retrieved it with the chemical queries. As the Reviewer points out, some other compounds are also similar to known DUB inhibitors but, although they share common sub-structures, the overall similarity is moderate. As reported earlier, Data S1 now includes the chemical structures and SMILES representations of all Top25 compounds, as well as the chemical structure and Tanimoto similarity of their closest DUB inhibitor. Please, find below a summary of this table, and we refer the Reviewer to our answers to Questions 6 & 7 for further assessment of the novelty of our findings, beyond mere similarity to DUB inhibitors.

1	Rank	SourceID	Chemical_query	Biological_query	InchIkey	SMILES	Structure	Structure closest DUB inhibitor	Similarity closest DUB inhibitor	PAINS	SureChEMBL	BMS	MLSMR	Dundee	Inpharmatica	Glaxo	LINT	TOTAL_ALERTS
2	1	PubPort-100-06		1	1	A/EBDRMLKXWVZSQ-LHFFFAOYSA:FFJZ(C)1-c1ccccc1			0.45	0	0	0	1	0	0	0	0	1
3	2	PR819_20		0	0	ZXOBLNBVNRQVLC-LHFFFAOYSA:1N			1	0	2	0	2	2	2	2	3	13
4	3	Protein-385		1	0	KKZJGLLVKMTCM-LHFFFAOYSA:N			1	1	0	0	8	7	0	1	1	18
5	4	T5870534		1	0	QLVXYVHFVQZKUC-LHFFFAOYSA:1			0.43	0	0	0	0	0	0	0	0	0
6	5	Protein-434		0	0	YJQPYGSHQPGBLKGSXXXXSSA:K			0.13	0	1	0	2	5	1	0	1	10
7	6	OMN-3-0167		1	0	TWOOIMKXVRYNG-VKAYKQESA:1			0.29	0	1	2	0	2	1	0	3	9
8	7	Protein-438		0	1	GSVQUGOUKJHRC-VYKPYRYVSA:1			0.17	0	2	0	3	4	3	0	7	19

Figure R5: Screenshot of the new tab in Data S1, where we have included the extra information requested by the Reviewers, to make it more informative and easier to follow.

5. Dimensionality reduction of the ECFP to 128 dimensions is too much compression, this does not exploit the full potential of chemical fingerprints. This must be discussed.

A standard representation of ECFPs has 2048 dimensions (bits) and reducing it to 128, as the Reviewer points out, implies a substantial compression. However, we would like to note that ECFPs are binary (0/1), while each component in our

signatures is a real number (R), which in practice allows for a much larger value distribution. We do not claim that our chemical descriptor outcompetes a consolidated fingerprint like ECFP4, especially in terms of interpretability. By design, our compression is specifically focused on preserving “similarities” between compounds. For this specific application, our compressed descriptors yield very similar results to the original ECFP fingerprints.

In particular, to assess the ability of CC signatures of the A1 space (i.e. the “compressed” fingerprint) to recover pairs of similar and dissimilar molecules according to their ECFPs, we randomly selected 10,000 molecules and considered their 1, 5 and 10 nearest neighbors (in the ECFP space) as “positive” pairs. As “negative” pairs, we randomly sampled an equal number of molecules. The graph below shows the area under the ROC curve measuring the capacity of compressed (A1) signatures to classify positive/negative ECFP pairs with a simple cosine similarity measure. Please, note that the signature compression occurs in Sign2, where we impose the 128 bit vectors. As it can be seen, both A1 Sign2 and Sign3 are indeed able to recapitulate the chemical similarities observed when ECFPs are used, while keeping the length of the descriptor in 128 dimensions. An advantage of A1 signatures is that they have a unified format with the rest of CC signatures (A2-E5), meaning that they can be readily stacked to the rest of our signatures for “global” measures of similarity.

Figure R6. AUROCs showing the capacity of chemistry A1 signatures (Sign1, Sign2 and Sign3) to recapitulate the 1, 5 and 10 nearest neighbors according to ECFP4 descriptors for a sample of 10,000 randomly selected compounds.

We have included a sentence describing the capacity of the A1 signatures to recapitulate the more complex ECPF4 descriptors and summarize the results in Figure S4.

6. Only three compounds are found exclusively with bioactivity queries, so it is fundamental to better discuss them. Which are their structures (see also comment 2)? Do they differ a lot from the known DUB inhibitors? Are they large compounds? If yes, a comparison should be carried out with structural fingerprints suitable for large molecules to see if they might be found.

We thank the Reviewer for his/her comment, which uncovered an important error in the annotation of the specific queries that identified each compound in the set of tested molecules, including the Top25. We apologize for the mistake and have now corrected everything related to it in the text, Figure 4 and Data S1.

In answer to the Reviewer's comment, and as highlighted above, we have now included the chemical structures of each active compound and have compared them to their closest DUB inhibitor. All these data are reported in Data S1. As it can be seen in Figure R7, the three compounds identified exclusively with biological queries (i.e. 7, 16 and 24) are chemically unrelated to any of the 45 known DUB inhibitors.

Rank	SourceID	Chemical_query	Biological_query	Inchikey	SMILES	Structure	Structure closest DUB inhibitor	Similarity closest DUB inhibitor
7	Prestw-838		0	1	GSVQIUGOUKJHRC-YFKPBYRVSA-N	 Prestw-838	 Compound 55	0.17
16	Prestw-994		0	1	BKYPTRYDKTTJY-ZDUSSCGKSA-N	 Prestw-994	 85-37	0.13
24	AG-690/40696652		0	1	VFGMLIXXTWJQD-UHFFFAOYSA-N	 AG-690/40696652	 589-0296	0.32

Figure R7. Information in Data S1 about the three active compounds identified exclusively with biological queries showing that they are drug-like molecules and show very limited Tanimoto similarities with any known DUB inhibitor.

We also assessed whether there was any undetected bias in the size (molecular weight) or the drug-likeness (QED) of the compounds selected by the different queries. As Figure R8 shows, we found no substantial difference in either measure.

Figure R8. Molecular weight (MW) and quantitative estimate of drug-likeness (QED) of all the compounds experimentally tested for activity against Snail1, splitting them according to the queries that selected them (chemistry only in red, biological only in blue, both queries in lilac, and random in grey).

In a more general analysis, we also calculated the Tanimoto similarities and the corresponding relative rankings between the 222 selected compounds and the 45 known DUB inhibitors finding that, as expected, the compounds identified by chemical queries have higher similarities than those retrieved by the biological queries (Figure R9).

Figure R9. Tanimoto similarity and relative ranking of the 222 selected compounds against *Snail1* compared to the closest known DUB inhibitor.

We then calculated the distances of the 222 selected compounds according to their composed signatures to the three different queries used, namely Q1: chemical similarity to known DUB inhibitors, Q2: molecules targeting proteins related to *Snail1* function (e.g. TGFBR) and Q3: molecules triggering similar transcriptional responses to the genetic perturbation of the targets considered in Q2. Thus, Q1 is based on chemical signatures (A1+A4), while Q2 contains target and network signatures (B4+C3+C4+C5) and Q3 is based on transcriptional signatures (D1). Please note that, since we use distances, we rank higher molecules having lower distances to the queries.

Figure R10 shows how, as expected, the compounds selected only by the chemical query Q1 are upranked compared to those selected by biological queries only or by the overlap of the two. Q2 distances confirms the upranking of compounds selected only by biological queries, while Q3 seems to marginally uprank compounds selected by both queries.

Figure R10. Query distances of the selected compounds split by chemistry only (red) biological (blue) and the overlap between the two (in lilac).

Finally, Figure R11 shows how the chemical and biological queries are complementary and they all are able to identify compounds with activity against Snail1 with a 3-5 fold enrichment over a random selection.

Figure R11. Fold enrichment of compounds selected by chemical query in red (Q1), biological queries in blue (Q2 + Q3) and the overlap between the two in lilac with respect to random picks, based on their capacity to modulate Snail1 levels (Firefly:Renilla assay).

These analyses have been included in the Results section and summarized in Figure S13.

7. As control please report how many of the top compounds for Snail1 are found with a “classical virtual screening” of the PWCK and IRB libraries using the 45 known DUB compounds as queries and ECFP4 as standard structural fingerprint.

This is a good point, and we thank the Reviewer for bringing it up. First, we would like to stress that the main objective of our signature-based compounds selection was to create a small library of compounds enriched in molecules with the capacity to modulate Snail1 activity. We then tested the 222 compounds selected plus 188 compounds randomly picked to evaluate the validity of our strategy. Thus, the first effect of moving to an ECPF4-based selection is that the initial selection of compounds will be different, and we do not have experimentally tested most of them.

To address the Reviewer’s comment, we have now implemented a compound selection strategy based on the similarities between the 45 known DUB inhibitors and the compounds in the IRB and Prestwick libraries. We calculated all Tanimoto similarities between the >17k compounds in our library and the 45 annotated DUB inhibitors and we selected the 222 compounds with the highest similarities. Additionally, for each compound, we also computed the average similarity to the closest 5 DUB inhibitors and, again, selected the 222 with highest scores. The Figure R12 below shows the overlap between the different selections, revealing that all three approaches are indeed complementary.

Figure R12. Venn diagram showing the overlap between 222 compounds selected from the IRB+Prestwick libraries (>17k molecules) using chemical checker signatures ('our selection') and ECPF4-based strategies.

We then calculated the enrichment in the identification of Snail1 inhibitors achieved by each strategy as explained in the m/s. In brief, we considered as positive hits all those compounds able to decrease 1.5 times the levels of Snail1, and compared the hits found by each strategy to those found in random compounds. As it can be seen in Figure R13, the signatures-based approach achieved a ~3.5-fold enrichment, closely followed by the average score of compounds over the 5 closest DUB inhibitors (~2.8-fold) and the one that only considers the highest Tanimoto similarity with any DUB inhibitor (~1.7-fold). Overall, the three approaches show a significant capacity to enrich compound libraries for a given function but, as expected, the combination of chemical and biological signatures identifies additional compounds.

Figure R13. Fold enrichment of compounds selected by Signature-based (red), average distance to the closest 5 DUB inhibitors (Top5; green) and closest distance to any DUB inhibitor (Top1; blue) queries with respect to random picks, based on their capacity to modulate *Snail1* levels (Firefly:Renilla assay).

A whole sub-section describing these analyses has been included in the Online Methods and Figure S25 summarizes the results.

8. In the paragraph “Enhanced prediction capabilities compared to chemical descriptors” a model (SigAR) is compared with a set of fingerprints (ECFPs). Therefore, one cannot conclude that the better performance is due to the use of bioactivity fingerprints. To establish so, one should use the bioactivity signature as fingerprint within the models of the benchmark (or in general compare the two types of fingerprints with the same model). Furthermore, ECFPs are not the descriptors available in MoleculeNet, the authors should extend their benchmark to (at least) all available fingerprints in MoleculeNet to demonstrate that bioactivity fingerprints perform better or different than established fingerprints.

We fully agree with the points raised by the Reviewer, which we have now addressed thoroughly. Previously, we chose to compare CC signatures with a popular fingerprint such as ECFP. As a supervised ML algorithm, we chose random forest classifiers for both types of descriptors, since this ML algorithm is well-known to the community and performs robustly across many ML tasks. Therefore, CC was compared to ECFP “fairly” because the same ML algorithm was used in both cases. To remove the effect of hyperparameters on performance, we used 20 iterations of a Bayesian Hyperparameter Optimization (HyperOpt) for every model and for each type of descriptor. Multiple train:test splits were done (5-fold cross-validations, performed in triplicates), in addition to a 10-fold Mondrian cross-conformal adaptation to estimate confidence. Overall, we put a substantial amount of computational resources into ensuring robustness of the classifiers, with the hope that this would let us focus on the difference between descriptors (CC-vs-ECFP) rather than the effect

of the ML algorithm of choice. We agree with the Reviewer that our previous efforts may not have been sufficient.

To address the Reviewer's concern, we have now extended our collection of "chemical" descriptors beyond ECFPs. In particular, we include Daylight-like (RDKit) fingerprints (path-based), MACCS keys and a data-driven state-of-the-art descriptor named CDDD, which is based on a deep-learning model trained on string (SMILES) representations of the molecules¹. Graph-based "layers" (e.g. graph convolutions from the molecule structure) are typically trained as part of the supervised ML task itself and therefore not usually available as generalistic descriptors. Thus, we do not include them in our internal benchmarks, as they are not natural inputs for a conventional classifier such as a random forest. Nonetheless, we report in Figure 5 the top performances in the MoleculeNet publication, which in some occasions included graph-based methods.

In the following Figure R14, we show the relative performance across the MoleculeNet of random forest classifiers trained on CC signatures with respect to the 4 chemistry-based fingerprints (CDDD, MACCS, ECFP, RDKit). To ease comparison, performance measures were scaled per task with a z-score (across the 4+1 descriptor types).

Figure R14. Relative performance of CC descriptors across the MoleculeNet. A z-score is calculated for each task based on the performance scores for that particular task across descriptors. For computationally intensive tasks (e.g. PCBA) we subsampled 50 tasks randomly. Please note that, in order to accommodate a comment raised by Reviewer #2, here CC refers to a fully stacked CC signature (25x128=3,200 dimensions). The prediction tasks are sorted according to the median of the CC performance.

Finally, and to confirm that the observed trends are not dependent on our choice to use a random forest classifier, we repeated the experiment with a model-agnostic

¹ The Continuous and Data-Driven Descriptors (CDDD) are chemical descriptors that were generated by training an Auto-encoder with a huge corpus of chemical structures. By encoding and decoding chemical structures, the model learned to compress the meaningful information into a continuous low-dimensional representation vector. The trained model is publicly available and can be used to generate descriptors for any molecule of interest.

approach based on the AutoML TPOT methodology. In brief, TPOT automatically performs feature selection / processing, classifier choice and hyperparameter optimization across a wide array of standard ML techniques. Thus, this approach provides a fair (ML-independent) head-to-head comparison between our descriptors and the rest of chemical fingerprints. Figure R15 shows that results are consistent with our previously observed trends with random forest classifiers. Note that, in this case, we could not address the PCBA MoleculeNet subtasks (involving 400k molecules) due to prohibitive computational costs for TPOT.

Figure R15. Analogous to Figure R14 with TPOT classifiers instead of random forest.

We have included a new paragraph in the Online Methods section and Figure S22 summarizes the results of using different chemical descriptors and classifiers.

9. Extending the description of the bioactivity fingerprint would increase the readability of the manuscript.

We thank the Reviewer for his/her comment, and the updated version of the m/s now includes a more detailed description of bioactivity fingerprints.

10. In figure 2g how is the chemical diversity assessed? It is unclear what low and high refer to.

True. Chemical diversity was assessed as follows. First, we used t-SNE to project CC GSigs from 3,200 dimensions to 2D. Then, for each point (molecule) in the map, we collected the 5 nearest neighbors calculated in the 2D space (i.e. the points in their vicinity). Finally, chemical “diversity” was calculated by averaging the Tanimoto similarity (ECFP4) between the molecule in question and their 5 neighbors. The blue-to-red scale refers to this average Tanimoto measure, ranging from 0 (chemically diverse) to 1 (chemically coherent). We now clarify it in the figure legend.

11. Figure 2h, legend. Is the cluster enriched with heat shock protein 90 (HSP90AA1) ...” I assume a word is missing, inhibitors perhaps? And molecules 4 and 5 are a good example of similarity/dissimilarity to which compound/ set of compounds?

We thank the Reviewer for pointing out the missing word. We have corrected it in the text. In addition, we have clarified that compound 4 is chemically similar to the surrounding points (5 nearest neighbors) in the map, whereas compound 5 is dissimilar to compounds in its vicinity.

12. Figure 3. How does the chemical space look like when using structural fingerprints? Do compounds still cluster by target? In these cases, is the awareness of the target coming from the bioactivity fingerprint or is it intrinsic in the chemical structure?

This is indeed a key point that we address in Figure S10 (see Figure R16 below), where we show the t-SNE 2D projections of the compounds in the Drug Repurposing Hub based on our GSigs and Morgan Fingerprints (ECFP4). We can appreciate that the ECFP4 does not achieve the same degree of clustering for drugs sharing the same mechanisms of action (MoA), at least for the highlighted cases (Figure S10a). A more systematic analysis confirmed this trend (Figure S10b). In particular, when we investigated all of the annotation criteria of the Drug Repurposing Hub (i.e. Target, MoA, therapeutic indication and disease), we observed that, in general, molecules belonging to the same category tend to be closer to each other in the GSig space compared to the ECFP4 space ($\Delta(CC, ECFP4) > 0$). In other words, the clustering achieved with GSigs is more relevant to the drug-related annotations than the clustering achieved with MFps. Thus, and answering the Reviewer's question, the bioactivity descriptors contain extra information that is useful to organize the chemical space.

Figure R16 (S10 in the m/s). Drug Repurposing Hub 2D projections. (a) *t*-SNE 2D projections based on GSigs (left) compared to Morgan fingerprint (MFp; 2048-bit, radius: 2) projections (right). Regions corresponding to certain MoAs are highlighted. (b) Level of clustering of the different annotations specified in the Drug Repurposing Hub, namely 'targets', 'MoA', 'indications' and 'disease areas'. Each dot corresponds to an annotation, and the size of the dot is proportional to the number of molecules. The average Euclidean distance in the 2D-projection between molecules with the same annotation is calculated, both for GSig- and MFp-based projections. For each annotation size, 100 randomly sampled points are drawn from the projection in order to scale the average distance measure. The x-axis measures the difference between GSig and MFp distances. Values close to 1 indicate that molecules of a certain annotation are well localized in the GSig projection and scattered in the MFp projection. Values close to -1 indicate the contrary.

Reviewer#2:

Bertoni et al describe a new approach for the prediction of bioactivity for small organic compounds. While existing methods in the field mostly focus on chemical descriptor like fingerprints, the authors try to capture all different types of data from chemical structure to known bioactivity data. Since data is sparse for many descriptor types, they use DNNs to predict missing values. For validation, the authors compare their approach named Chemical Checker (CC) to a series of prediction tasks defined in DeepChem. Furthermore, they apply it Snail1 and report new chemical matter found.

In general, machine learning for bioactivity prediction is a very hot topic which is not surprising given the advances in ML and the relevance of the prediction task for drug discovery. The study presented has - to the best of my knowledge - a new element, namely the integration of heterogeneous data into one system. It has been shown before, that transfer learning is helpful and maybe one of the key strength of DNNs in cases of sparse data like we have in early drug discovery. Therefore, the manuscript content deserves publication although the gain in prediction quality is in my view more incremental than ground breaking. I recommend the authors to address several issues before publication:

We would like to thank the Reviewer for assessing our m/s and, indeed, for acknowledging the importance of having new bioactivity descriptors to train ML models and recommending its publication. We address in detail his/her comments below.

1- In the introduction, the authors mostly ignore the whole field of chemogenomics. There are numerous approaches based on the idea of predicting bioactivity from bioactivity data. The first, to my knowledge, is from Kauvar in 1995 (Chem.Biol.) which used bioactivity data as fingerprints. Mostly known is certainly the SEA approach (Keiser et al, Nature 2009). Of course, these papers have different focus, but they are of importance to put this work into the right scientific context.

The Reviewer is indeed right, and we apologize for having omitted many of the pioneering work in the development of small molecules bioactivity descriptors. We cited the most relevant papers in the presentation of the Chemical Checker (Duran-Frigola et al. Nat Biotechnol 2020), but overlooked in the first version of this manuscript. We have now corrected it and cite some of the most relevant literature, including the two papers suggested by the Reviewer.

2- The whole structure of the manuscript makes it very hard to follow. Almost all details to really understand the methods are not where they are needed. The manuscript would greatly benefit from a clear method description including important parametrization. Since the authors publish source code, this might not hinder reproducibility, but it certainly hinders to get a good understanding of the many technical elements of this work.

We thank the Reviewer for his/her comment and, in this revised version of the m/s, we have added some extra methodological explanations in the main text but, to preserve its flow, we left the most technical details for the Online Methods section

and, as the Reviewer points out, the source code with all the parametrization is publicly available.

3- The technical validation based on retrospective data needs substantial improvement. Complex machine learning methods like the presented one should be validated more critically, for example by y-scrambling experiments, by showing the performance delta by skipping individual steps of the method, by cluster-cross validation or time-series analysis.

We thank the Reviewer for his/her comment and suggestion.

First of all, we would like to clarify the development of “new” classifiers (i.e. supervised ML for MoleculeNet) is, we feel, beyond the scope of the paper. Rather, our goal is to present a new set of small molecule bioactivity descriptors that can be readily used across a wide range of ML algorithms, and that are complementary to the currently available chemical descriptors. Indeed, we limit our study to the use of Random Forest classifiers (which are a standard, well-accepted solution), and compare their performance when different molecular descriptors are used. We are convinced that a better selection and training of ML algorithms specific for each task could significantly increase the performances achieved, but it is not the objective of the current manuscript.

Having clarified this point, we agree with the Reviewer that the paper could benefit from some additional validations. We have performed the tests suggested for the central asset of this paper, i.e. the inferred CC signatures (GSig).

On the one hand, as suggested by the Reviewer, we conducted a scrambling experiment. In the context of GSigs, we suggest that a compelling scrambling test would involve the calculation of GSigs related to randomized MFps, in order to assess whether the resulting GSigs would have some sort of “structure” (signal) or not. This is, we randomly picked 1,000 compounds from the CC universe and computed their global signatures (GSig) with and without a scrambling of their structural representation (i.e. their Morgan fingerprints). Then, we calculated all the pairwise distances signatures in each set. As expected, scrambled signatures are much more similar between them and thus, when compared to GSigs, also show shorter Euclidean or cosine distances than the intra-GSig signature comparisons, which maintain the desired variability (Figure R17). Indeed, and also as expected, when we plot only the shortest distance between pairs of molecules within the GSig set and scrambled-GSig, we see how there are no significant similarities between scrambled and global signatures, while GSigs can indeed detect similarities between compounds - i.e. short distances between GSigs (Figure R18). Finally, 2D projections (t-SNE and PCA) of scrambled and global signatures (Figure R19) reinforce the observation that scrambled signatures are indeed different from real GSigs and contain no bioactivity signal.

Figure R17. Distribution of all-against-all Euclidean and cosine distances between GSigs (green) and between scrambled and GSigs (orange).

Figure R18. Distribution of the closest Euclidean and cosine distances between pairs of molecules within the GSig set (green) and scrambled-GSig (orange). The plots show how there are no significant similarities between scrambles and global signatures, while GSigs can indeed detect similarities between small molecules (i.e. short distances between GSigs).

Figure R19. 2D projections (t-SNE and PCA) of scrambled and global signatures (GSig).

On the other hand, during the revision process we updated the Chemical Checker (CC) resource, which gave us the opportunity to perform an interesting time-series analysis, as suggested by the Reviewer. More specifically, we confronted the small molecule bioactivity signatures predicted using the 2019_01 signaturizers (presented in this paper) with novel experimental bioactivity data available in the newly released 2020_02 version. In this new CC release, most of the source databases have been updated presenting new data for molecules that were not present in the previous releases.

Profiting from the novel experimental data, for each bioactivity space we gathered molecules only present in the new version and that are completely novel in terms of CC annotation. Using the signaturizer 2019_01 we predicted signatures for the new molecules and searched for neighbors in the 2019_01 CC universe. We then compared this set of neighbors to those confirmed in the 2020_02 version, excluding molecules not available in the 2019_01 CC universe (i.e. those neighbors that are also new molecules).

Figure R20 shows the fraction of novel molecules (y-axis) for which at least one correct neighbor is identified among the top [1-1,000] predicted neighbors (x-axis), within the roughly 1M molecules in the CC chemical space. We also show the same fraction of recovered real neighbors when randomizing the 'signaturized' molecules. Despite the limited number of new molecules with experimental information in some of the spaces, we can see that the signaturizers derived from previous versions can identify similar molecules for a significant fraction of the new compounds. Moreover, we also see how increasing the applicability score threshold augments the reliability of the predicted signatures in all the bioactivity spaces. However, as expected, when we use random molecules as bait, we cannot identify true neighbors and there is no relationship with the applicability scores.

Figure R20. Time-series validation of the signaturizers in all bioactivity spaces. The plots show the ability of the signaturizers to identify similar molecules to compounds not yet annotated when the signaturizers were derived, and we compare them to a random expectation. Despite the limited number of new molecules with experimental information in some of the spaces, we can see that the signaturizers derived from previous versions can identify similar molecules for a significant fraction of the new compounds, and this increases when we use more stringent applicability thresholds.

Finally, we kindly refer this Reviewer to our answer to question 8 by Reviewer #1, where we explain additional validations to the MoleculeNet supervised learning exercise, including exploration of more chemical fingerprints and application of automated ML techniques (TPOT). Our descriptors perform systematically better than any of the chemical fingerprints.

We have added a paragraph in the Results section and a complete new sub-section in the Online Methods describing the additional validations performed, and the results are summarized in Figures S23 and S24.

4- The comparison to DeepChem predictions is technically not sound. I have never seen that descriptors are taken out instead of similar data points. For comparison, all molecules similar to the testset compounds have to be removed from all datasets used for training, otherwise the experiment is inconclusive. Furthermore, for comparison the best performing chemical descriptors which are graph descriptors rather than simple morgan fingerprints have to be taken into account.

We understand the criticism of the Reviewer and apologize because our writing was not clear enough. Indeed, we followed MoleculeNet recommendations strictly, both in terms of splitting (e.g. scaffold-based) and in terms of performance measure (AUROC/AUPR) to ensure a fair assessment. This is now clarified in the text.

As for our decision to “take out” descriptors, we actually did this with our good will to provide the fairest-possible scenario for comparison between the CC and MFps, removing data types that may be directly or indirectly related to the prediction task in question. In fact, it is clear that taking out descriptors is detrimental for the CC (Figure 5). Even in this challenging scenario, the CC outperforms the chemical fingerprints.

Finally, we agree with the Reviewer that graph convolutional layers achieve very good performances in supervised prediction tasks like the ones available in MoleculeNet. Indeed, performance of graph-based classifiers is accounted for in Figure 5 when we show the “best MoleculeNet” scores. Following a related comment by Reviewer #1, and as mentioned above, we expanded the collection of chemical descriptors to include other classical options (MACCS, Daylight-like, ECFP4) as well as more modern descriptors based on deep learning; namely, the CDDD. Strictly speaking, CDDD is not a graph-based fingerprint. Rather, it is trained on string-based representations of the molecules (e.g. SMILES) over a large corpus, but it has the advantage of being conveniently expressed in a vectorial format, which makes it an amenable input for a conventional ML algorithm such as a random forest classifier. Our priority was to “remove” the effect of the classifier in our comparison between GSigs and chemical representations. Therefore, we trained all models with the same ML algorithm (random forest). The most popular graph-based methods involve a specific type of deep (convolutional) neural network classifier and therefore are not fit for this comparative analysis.

Overall, we carried out a substantial amount of additional training cycles, adding 4 new fingerprints and model-agnostic validations beyond random forest classifiers (Figure R14, R15). Considering the number of cross-validation splits, hyperparameter optimization iterations, splits for the Mondrian cross-conformal predictions, and individual training for each MoleculeNet subtask, this represents a considerable computational effort and, we hope, provides sufficient and systematic evidence of the added value of GSigs with respect to chemistry-based descriptors.

As described above, we have added multiple references in the text and a complete new sub-section in the Online Methods to describe our validations in detail.

5- For the Snail1 experiment, the presented data is insufficient. The data tables contain InChI Keys rather than SMILES or another readable format making it extremely difficult to look at the molecules. There is no PAINS filter or frequent hitter analysis. There is no analysis of the similarity of the new compounds to known Snail1 inhibitors, no comparison of bioactivity levels. Furthermore, specialized journals like JMedChem, etc. do require dose response curves for all active compounds.

We thank the Reviewer for his/her comment, which will certainly improve the readability of our m/s. We have now updated the Data S1 file to include SMILES for each tested compound. We have also included a number of structural alerts, including the well-known PAINS filters², the SureChEMBL Non-MedChem Friendly SMARTS (SureChEMBL)³, the Bristol-Myers Squibb HTS Deck filters (BMS)⁴, the University of Dundee NTD Screening Library Filters (Dundee)⁵, the NIH MLSMR Excluded Functionality filters (MLSMR), fragments derived by Inpharmatica Ltd. (Inpharmatica), the Pfizer lint filters (LINT)⁶, and the Glaxo Wellcome Hard filters (Glaxo)⁷. Additionally, and also in response to Reviewer #1, for the Top25 compounds we also include their structure as well as the structure and Tanimoto similarity to the closest known DUB inhibitor. As for a more classical chemical comparison and screens, we kindly refer this Reviewer to the analyses presented above.

Finally, we would like to stress that the main objective of the current work is to present a new set of bioactivity descriptors that can be derived for any uncharacterized molecule thanks to the signaturizers. We then used them to infer bioactivity signatures to our collection of mostly uncharacterized ~20k compounds, and sought to assess whether they are useful to select a much smaller collection enriched in small molecules that might be active against Snail1 activity. This is only a proof-of-principle exercise, and we do not claim to have found potent Snail1 (indirect) inhibitors. Our results show that, indeed, the chemical and biological queries are able to find roughly 3-5 more active compounds than a random selection, on a very simple single-dose (20 μ M) dual-luciferase assay. Then, for the top10 most active compounds, we tested three different concentrations (40, 20 and 10 μ M), without any attempt to identify the activity range, and found that five of them (1, 2, 4, 7, 9) displayed a dose-dependent response (Figure R21). We are aware that further analyses would be needed to identify potent Snail1 inhibitors beyond this initial screen, and we are following up on some of the molecules (see Figure R22 below). Although we feel that presenting these follow-up experiments is outside the scope of the current paper, and it could distract the attention from the bioactivity descriptors, we will be happy to include them if the Reviewer #2 deems it necessary.

Figure R21. Top10 activity compounds tested on the dual-luciferase assay with three doses (40, 20 and 10 μ M). We can appreciate that five of the compounds, namely 1, 2, 4, 7, 9 show a dose-dependent inhibition of the Snail1 activity. The bar plot shows the average and standard deviation over three independent experiments.

Figure R22. Initial validation of the Snail1 inhibitory activity of compounds #1 (Molport-009-100-062), #3 (Prestw-385) and #4 (T5870534) measuring the endogenous amount of Snail1 in MDA-MB-231 cells and also in cancer associated fibroblasts (VAFs). As it can be seen *cmp#1* and *cmp#4* show a clear dose-dependent activity in both cell types.

We have included Figure R21 above as Figure S14 in the m/s to show the complete experiment.

References

- 1 Lambies, G. *et al.* TGFbeta-Activated USP27X Deubiquitinase Regulates Cell Migration and Chemoresistance via Stabilization of Snail1. *Cancer Res* **79**, 33-46, doi:10.1158/0008-5472.CAN-18-0753 (2019).

- 2 Baell, J. B. & Holloway, G. A. New substructure filters for removal of pan assay interference compounds (PAINS) from screening libraries and for their exclusion in bioassays. *J Med Chem* **53**, 2719-2740, doi:10.1021/jm901137j (2010).
- 3 Sushko, I., Salmina, E., Potemkin, V. A., Poda, G. & Tetko, I. V. ToxAlerts: a Web server of structural alerts for toxic chemicals and compounds with potential adverse reactions. *J Chem Inf Model* **52**, 2310-2316, doi:10.1021/ci300245q (2012).
- 4 Pearce, B. C., Sofia, M. J., Good, A. C., Drexler, D. M. & Stock, D. A. An empirical process for the design of high-throughput screening deck filters. *J Chem Inf Model* **46**, 1060-1068, doi:10.1021/ci050504m (2006).
- 5 Brenk, R. *et al.* Lessons learnt from assembling screening libraries for drug discovery for neglected diseases. *ChemMedChem* **3**, 435-444, doi:10.1002/cmdc.200700139 (2008).
- 6 Blake, J. F. Identification and evaluation of molecular properties related to preclinical optimization and clinical fate. *Med Chem* **1**, 649-655, doi:10.2174/157340605774598081 (2005).
- 7 Hann, M. *et al.* Strategic pooling of compounds for high-throughput screening. *J Chem Inf Comput Sci* **39**, 897-902, doi:10.1021/ci990423o (1999).

Reviewers' Comments:

Reviewer #1:

Remarks to the Author:

The authors have addressed my comments and request with very thorough revisions and provided additional data that resolves the open questions, in particular regarding the validity of their performance comparison between classical fingerprints and their bioactivity fingerprint, reporting compound structures, and checking that there was no artifact by excluding luciferase inhibition. The revised manuscript is suitable for publication.

Reviewer #2:

Remarks to the Author:

This reviewer appreciates the additional work the authors spent into this manuscript. However, I have to admit that my general impression remains. While many information is given in various plots which this reviewer does not find helpful in really evaluating the performance of the method, relevant information is mostly hidden in the supplementary or external data files (like Data S1). While the authors do not show full dose-response curves, they mention 5 molecules with dose-dependent behavior. These molecules are, however, all found by chemical queries. Most of them show high similarity to DUB inhibitors. The authors added more data with respect to the MoleculeNet comparison, the difference in performance remains incremental. Two minor comments: The introduction now cites two more papers with respect to bioactivity descriptors, but it fails to clearly describe the methodological achievements of this paper against the state of the art. Y-scrambling is a standardized statistical procedure scrambling the outcome, not the descriptor.

Point-by-point responses to Reviewer's comments

NCOMMS-20-38745-T "Bioactivity descriptors for uncharacterized chemical compounds" by Bertoni*, Duran-Frigola*, Badia-i-Mompel*, et al.

We would like to thank the Editor and Reviewers for their critical reading of our manuscript and the positive feedback, accepting it for publication in Nat Commun. As indicated by the Editor, we have now addressed in the minor issue raised by Reviewer#2, and have highlighted the changes in the manuscript and Supplementary Information in **yellow**.

Please find below a point-by-point response to Reviewers' comments.

Reviewer #1 (Remarks to the Author):

The authors have addressed my comments and request with very thorough revisions and provided additional data that resolves the open questions, in particular regarding the validity of their performance comparison between classical fingerprints and their bioactivity fingerprint, reporting compound structures, and checking that there was no artifact by excluding luciferase inhibition. The revised manuscript is suitable for publication.

We thank the Reviewer for his/her recommendation to accept our m/s for publication.

Reviewer #2 (Remarks to the Author):

This reviewer appreciates the additional work the authors spent into this manuscript. However, I have to admit that my general impression remains. While many information is given in various plots which this reviewer does not find helpful in really evaluating the performance of the method, relevant information is mostly hidden in the supplementary or external data files (like Data S1). While the authors do not show full dose-response curves, they mention 5 molecules with dose-dependent behavior. These molecules are, however, all found by chemical queries. Most of them show high similarity to DUB inhibitors. The authors added more data with respect to the MoleculeNet comparison, the difference in performance remains incremental.

We thank the Reviewer for his/her comments and positive feedback. And have addressed the minor comments highlighted.

Two minor comments: The introduction now cites two more papers with respect to bioactivity descriptors, but it fails to clearly describe the methodological achievements of this paper against the state of the art.

We thank the Reviewer for his/her comment. However, in our opinion, most of the Discussion section is devoted to highlight the added value of our bioactivity descriptors, their complementarity with current available chemistry descriptors and also their limitations. We do not see how we could stress these points more clearly without being repetitive and without entering in detail about other descriptors which, we think, it is beyond the scope of our paper.

Y-scrambling is a standardized statistical procedure scrambling the outcome, not the descriptor.

The reviewer is right in that Y-scrambling is a procedure to randomize the outcome, not the descriptor. We now understand that this Reviewer, in the first round of comments, was referring to the MoleculeNet benchmark experiment. Initially, we thought he/she was asking us to do Y-scrambling on the siamese neural network (signaturizer) experiment; which was not obvious since, in that case, the output is an inner layer of the network corresponding to a descriptor indeed. This is why we did a non-standard adaptation of Y-scrambling to our best.

We have now performed the more standard Y-scrambling across MoleculeNet. In particular, we performed outcome randomisations on the whole array of MoleculeNet prediction tasks (previously introduced). Y-scrambling (or Y-randomization) is commonly used to validate QSPR/QSAR models by training predictors on the randomly shuffled dependent variable (Y) that we aim to predict, as pointed out by the reviewer. Comparing the performances (on an unshuffled test set) of models trained on shuffled versus unshuffled data we can assess the robustness of descriptors and rule out the effect of random correlations. This is exactly what we observe for both GSig and ECPF4 descriptors (Supplementary Fig 24), with a drop of model performances to an average of 0.5 ROC AUC when training on scrambled data.

Supplementary Figure 24. Y-scrambling experiment on MoleculeNet task. We performed a simple logistic regression using both GSig signatures (CC in blue) signature and ECPF4 descriptors (MFP in red). The box in dark shades corresponds to normal classification on unscrambled Y data, while the light shades correspond to 5 repetition of scrambling Y data.